



# Country-level estimates of gross and net carbon fluxes from land use, land-use change and forestry

Wolfgang A. Obermeier[1], Clemens Schwingshackl[1], Ana Bastos[2], Giulia Conchedda[3], Thomas Gasser[4], Giacomo Grassi[5], Richard A. Houghton[6], Francesco N. Tubiello[3], Stephen Sitch[7], and Julia Pongratz[1,8]

[1]Ludwig Maximilians University Munich, Munich, Germany
[2]Max Planck Institute for Biogeochemistry, Jena, Germany
[3]Food and Agriculture Organization of the United Nations, Rome, Italy
[4]International Institute for Applied Systems Analysis, Laxenburg, Austria
[5]Joint Research Centre of the European Commission, Ispra, Italy
[6]Woodwell Climate Research Center, Falmouth, USA
[7]University of Exeter, Exeter, United Kingdom
[8]Max Planck Institute for Meteorology, Hamburg, Germany

**Correspondence:** Wolfgang A. Obermeier (wolfgang.obermeier@lmu.de)

**Abstract.**

The reduction of $CO_2$ emissions and the enhancement of $CO_2$ removals related to land use are considered essential for future pathways towards net-zero emissions and mitigating climate change. With the growing pressure under global climate treaties, country-level land use $CO_2$ flux data are becoming increasingly important. So far, country-level estimates are mainly

available through official country reports, such as the greenhouse gas inventories reported to the United Nations Framework Convention on Climate Change (UNFCCC). Recently, different modelling approaches, namely dynamic global vegetation models (DGVMs) and bookkeeping models, have moved to higher spatial resolutions, which makes it possible to obtain model-based country-level estimates that are globally consistent in their methodology. To progress towards a largely independent assessment of country reports using models, we analyse the robustness of country-level $CO_2$ flux estimates from different

modelling approaches in the period 1950–2021 and compare them with estimates from country reports.

Our results highlight the general ability of modelling approaches to estimate land use $CO_2$ fluxes at the country-level and on higher spatial resolution. Modelled land use $CO_2$ flux estimates generally agree well, but the investigation of multiple DGVMs and bookkeeping models reveals that the robustness of their estimates strongly varies across countries, and substantial uncertainties remain even for top emitters. Similarly, modelled land use $CO_2$ flux estimates and country report-based estimates

agree reasonably well in many countries once their differing definitions are accounted for, although differences remain in some other countries. A separate analysis of $CO_2$ emissions and removals from land use using bookkeeping models also shows that historical peaks in net fluxes stem from emission peaks in most countries, whereas the long-term trends are more connected to removal dynamics. The ratio of the net flux to the sum of $CO_2$ emissions and removals from land use (the net-to-gross flux ratio) underlines the spatio-temporal heterogeneity in the drivers of net land use $CO_2$ flux trends. In many tropical regions,

net-to-gross flux ratios of about 50% are due to much larger emissions than removals; in many temperate countries, ratios close to zero show that emissions and removals largely offset each other. Considering only the net flux thus potentially masks



large emissions and removals and the different timescales upon which they act, particularly if averaged over countries or larger regions, highlighting the need for future studies to focus more on the gross fluxes.

Data from this study are openly available via the Zenodo portal at https://doi.org/10.5281/zenodo.8144174 (Obermeier et al., 2023).

# 1 Introduction

The carbon dioxide ($CO_2$) flux from land use, land-use change and forestry (LULUCF) is a key component of the global carbon cycle (Pongratz et al., 2021), and the net $CO_2$ flux from LULUCF ($f_{LULUCF}$) contributed about 10 – 15% to the total anthropogenic $CO_2$ emissions in recent decades (Friedlingstein et al., 2022b). Under the efforts towards achieving the long-term temperature targets formulated in the Paris Agreement, the importance of $f_{LULUCF}$ is expected to increase in the future (Fuss et al., 2018), potentially contributing ∼30% of the global climate change mitigation needed in 2050 to reach the 1.5°C target via both emission reduction and $CO_2$ removal (Roe et al., 2019). Despite its outstanding importance, estimates of $f_{LULUCF}$ remain subject to high uncertainty. For example, within the annual Global Carbon Budget (GCB) assessments of the Global Carbon Project, the net $f_{LULUCF}$ has the highest relative uncertainty among all terms (∼60% for most recent decade; see Friedlingstein et al., 2022b for GCB2022).

At the global scale, such as in the assessments of the Intergovernmental Panel on Climate Change (IPCC) and the GCB, $f_{LULUCF}$ is usually assessed by modelling approaches (Friedlingstein et al., 2022b), namely semi-empirical bookkeeping models (BKs) and dynamic global vegetation models (DGVMs), and more recently by merging bottom-up inventory estimates built up from country-level information (Grassi et al., 2018, 2023). Global modelling approaches have the advantage of being globally consistent, thus enabling, for example, the analysis of the global carbon cycle. Given recent improvements in global modelling approaches, such as better representation of processes related to land management, vegetation physiology, and soil biogeochemistry as well as increased spatial resolutions (Arneth et al., 2017; Blyth et al., 2021; Pongratz et al., 2021), several analyses of $f_{LULUCF}$ estimates using multiple approaches, including models, at country (e.g. Federici et al., 2017; Rosan et al., 2021; Schwingshackl et al., 2022; Grassi et al., 2023) and regional scales (e.g. Bastos et al., 2020b; Petrescu et al., 2021; Tubiello et al., 2021b; Nabuurs et al., 2022) have recently been conducted. Yet, their results might be less robust and consistent at finer spatial scales. Bottom-up approaches, such as national greenhouse gas inventories (NGHGIs), quantify $f_{LULUCF}$ based on inventory data that countries regularly submit to the United Nations Framework Convention on Climate Change (UNFCCC). Even though NGHGIs should fulfill best practice guidance from the IPCC (IPCC, 2006), the quality, methodological complexity, and provision of data used by the reporting countries vary (Grassi et al., 2021, 2022). Additionally, the Food and Agriculture Organization of the United Nations (FAO) disseminates independent bottom up estimates of LULUCF emissions and removals, based on forest area and carbon stock information provided every five years by the countries to the FAO Global Forest Resources Assessment (FRA), complemented by geospatial information on biomass fires and peatland degradation (Tubiello et al., 2022).





Land-based climate change mitigation can be achieved via two levers, namely by decreasing gross emissions and by increasing gross removals (the latter is often referred to as negative emissions, e.g. Fuss et al., 2018). These gross fluxes, which add up to the net $f_{LULUCF}$, act on different timescales and are in reality often linked by common land use practices (note that throughout the manuscript the term "$CO_2$ fluxes" generally refers to anthropogenic fluxes from land use). For example, if fewer wood products are harvested in forestry, this quickly leads to lower emissions but also to lower forest regrowth and thus lower removals, balancing towards net-zero fluxes in the longer-term (Gasser et al., 2022). In this study, we define gross fluxes as the sum of all fluxes related to certain LULUCF practices that typically lead to emissions or removals, respectively, in line with the Global Carbon Project. Gross emissions from LULUCF ($CO_2$ fluxes from the biosphere to the atmosphere, i.e. C source) are mainly related to cropland or pasture expansion resulting in the destruction of natural ecosystems by deforestation, forest and peatland degradation, as well as land use practices, such as biomass burning or forest management causing the decay of harvested wood products (HWPs, Friedlingstein et al., 2022b). Gross removals from LULUCF ($CO_2$ fluxes from the atmosphere to the biosphere, i.e. C sinks) are associated with land use changes such as afforestation and reforestation including the regrowth of secondary forest after agricultural abandonment, as well as land use influences such as forestry cycles and restoration of other (non-forest) ecosystems. Noteworthy, the $CO_2$ removal potential of LULUCF is deemed the most suitable and easily scalable option for negative emissions, potentially providing 25% of net greenhouse gas emissions reductions by 2030, primarily via afforestation, reforestation and management of existing forests (Griscom et al., 2017; Gidden et al., 2022; Smith et al., 2023). This is reflected by the fact that negative emissions from LULUCF are already widely included in the nationally determined contributions (NDCs) for climate change mitigation (Grassi et al., 2017; Smith et al., 2023). While this clearly highlights the importance of separately estimating gross emissions and gross removals, global assessments of $f_{LULUCF}$ usually consider net $f_{LULUCF}$ only, and, as stated by Houghton (2020), little attention has been paid to its components. While some studies explicitly separated gross fluxes early on (Tubiello et al., 2015; Federici et al., 2015), GCB studies considered gross fluxes for the first time in 2020 (Friedlingstein et al., 2020), with the most recent GCB2022 estimating global anthropogenic gross emissions at $3.8 \pm 0.7$ GtC yr$^{-1}$ and gross removals at $2.6 \pm 0.4$ GtC yr$^{-1}$ for 2012 – 2021, being thus 2 – 4 times larger than global net $f_{LULUCF}$ (Friedlingstein et al., 2022b).

The aim of this study is to provide a comprehensive national and regional comparison that integrates the different approaches and definitions around the world and complements previous studies on selective countries or regions. We investigate the robustness of net $f_{LULUCF}$ estimates from models, namely from BKs and DGVMs, we provide estimates at country- and regional-level and, additionally, we compare net estimates from models and inventories. To identify the drivers of the spatio-temporally varying net $f_{LULUCF}$ estimations from these assessments, we present and discuss country-level gross $f_{LULUCF}$ estimates as well as the net-to-gross flux ratio from BKs in addition to net flux estimates. This allows us a more nuanced analysis and identification of the land component processes that are relevant in specific regions and provides a quantification of land-based climate change mitigation potentials distinctly for the two levers "reducing emissions" and "increasing removals". The need for such an assessment is underlined by the increasing political relevance of LULUCF fluxes for countries' emissions reduction pledges, for example, in support of the ongoing Global Stocktake and the Glasgow Leaders' Declaration on Forests and Land Use of the





26th UN Climate Change Conference of the Parties (COP26) in Glasgow.

## 2 Overview of different $f_{\text{LULUCF}}$ assessment methods

We use data from BK and DGVM models, NGHGIs, and FAOSTAT for our analysis. The approaches are briefly described below and explained in more detail in appendix A.

### 2.1 Bookkeeping models (BKs)

BKs are explicitly designed to estimate LULUCF fluxes following land management and land use changes by tracking the carbon content in soil, vegetation and product pools, i.e. stocks and fluxes of carbon in the land biosphere and between land and atmosphere. They combine spatial information on land use activities with observation-based carbon stock densities and specific response curves of soil and vegetation carbon for each land-use conversion type (for more details refer to Sect. A1 and Pongratz et al., 2014). Using separate response curves for carbon release (emissions) and carbon uptake (removals), BKs model both gross fluxes explicitly, and net $f_{\text{LULUCF}}$ is derived as the sum of the two gross fluxes. BKs do not model the fluxes from peatland fire and drainage but these emissions are added on top of the BK estimates in this study according to the approaches used in the GCBs, compare Sect. A1. We use simulations by three BKs that were conducted for the GCB2022 (Friedlingstein et al., 2022b), namely 1) the 'Bookkeeping of Land Use Emissions' model (hereafter BLUE22; Hansis et al., 2015), 2) the 'Houghton and Nassikas' model (hereafter H&N22; Houghton and Nassikas, 2017), and 3) the compact Earth system model 'OSCAR' (hereafter OSCAR22; Gasser et al., 2020).

### 2.2 Dynamic Global Vegetation Models (DGVMs)

DGVMs are process-based models used to simulate the interaction between land surface and vegetation processes with the atmosphere. They approximate net $f_{\text{LULUCF}}$ as the difference in net biome productivity (NBP) of a simulation including LU-LUCF (similarly to the BKs, using spatial information on land use activities) and a simulation excluding LULUCF (the latter using a time-invariant pre-industrial land use map; for more details refer to Sect. A2 and Obermeier et al., 2021). DGVM simulations using transient (observed) environmental forcing are operationally available and can used to estimate the impact of climate variability or long-term environmental changes on $f_{\text{LULUCF}}$. In addition, DGVM simulations can be forced with constant environmental data prescribing, for instance, pre-industrial or present-day environmental conditions. Simulations using the latter setup most closely resemble conditions that occurred during the time when observed carbon densities (as used by BKs or inventories) were measured, and are therefore the best DGVM setup to compare with BKs or inventories. We here use nine DGVMs that provided simulations with present-day as well as transient environmental forcing within the project "Trends and drivers of the regional-scale emissions and removals of carbon dioxide" (TRENDY; Le Quéré et al. (2014); Sitch et al. (2015), Sitch et al., 2023 *in prep*).



## 2.3 National Greenhouse Gas Inventories (NGHGIs)

NGHGIs are the official country reports to UNFCCC that include estimates of greenhouse gas emissions and removals from
LULUCF. They widely rely on empirical emission factors in combination with country-level data on land use activities to
estimate $f_{\text{LULUCF}}$ (for more details refer to Sect. A3 and Grassi et al., 2022). The methods and report details strongly vary
between and among non-Annex I and Annex I countries. Non-Annex I countries frequently use default IPCC emission factors
(IPCC, 2006) and often only report fluxes from deforestation, while estimates from Annex I countries are often based on
country-specific statistical or process-based models for all IPCC land use categories (forest land, cropland, grassland, wetlands,
settlements and other land). For the analysis presented here, we use the country database (DB) compiled by Grassi et al. (2022),
as updated in Grassi et al. (2023) (hereafter referred to as NGHGI DB), which includes GHG data from all available country
reports submitted to UNFCCC, gap-filled when necessary to allow a complete time series from 2000 until 2020. According
to UNFCCC guidelines, country reports should encompass all LULUCF fluxes from any areas considered managed and for
which IPCC provide methodological guidance. In most cases, NGHGIs include natural and indirect anthropogenic fluxes (e.g.,
from $CO_2$ fertilization; Grassi et al., 2018; Schwingshackl et al., 2022; IPCC, 2010). We adjust the NGHGI DB data to match
the processes and definitions of the modelled estimates (the basis in this study) to correct for this so-called managed land issue
(hereafter, adjusted NGHGI DB), by subtracting the natural and indirect effects (i.e., human-induced environmental change)
from managed land. These effects are estimated by the ensemble mean of 16 transient DGVM simulations without land-use
changes from TRENDYv11 (corresponding to the 'natural terrestrial sink' in recent GCBs of the GCP), except for Brazil and
Canada filtered with an intact/non-intact forest mask (Potapov et al., 2017), according to the approach described in Grassi et al.
(2023). For Brazil and Canada, this approach uses the national gridded maps on managed and unmanaged forests used in the
respective NGHGIs (Brazil, 2020; Canada, 2021).

## 2.4 Statistical Division of the Food and Agricultural Organisation (FAOSTAT)

FAOSTAT $f_{\text{LULUCF}}$ estimates resemble the bottom-up approach of the UNFCCC data in the way that they are based on con-
sistent underlying activity data — at grid cell or at country level – in combination with emission factors. FAOSTAT $f_{\text{LULUCF}}$
data are estimated by applying IPCC Guidelines (IPCC, 2003, 2006) to activity data generated either through official country
reporting processes or through geospatial data analysed by FAO (for more details refer to Sect. A3 and FAO, 2020; Tubiello
et al., 2021b). FAOSTAT $f_{\text{LULUCF}}$ cover carbon emissions and carbon removals in forests and from deforestation, derived from
national carbon stock change statistics and IPCC emission factors, as well as emissions from peatland fires and peat drainage,
the latter two obtained from satellite imagery in combination with IPCC emission factors (Conchedda and Tubiello, 2020;
Tubiello et al., 2021b; Prosperi et al., 2020). The IPCC (2006, 2019) recognizes explicitly that both FAO activity data and
FAO emissions estimates provide a valuable set of reference data that can be used for validation, quality control, and quality
assurance of the data submitted through NGHGIs.



## 2.5 Main differences between the approaches

Although the different approaches summarized above (and described in detail in appendix A) all aim at quantifying $CO_2$ fluxes from LULUCF, they differ substantially. Some of the key differences between the approaches are briefly described below, further details are provided in the results and appendix section.

Uncertainties in $f_{\text{LULUCF}}$ estimations from modelling approaches mainly arise at high spatial resolutions (Kondo et al., 2022), from differences in underlying land use and land cover information (Gasser et al., 2020; Hartung et al., 2021; Ganzenmüller
et al., 2022), missing observational constraints (Goll et al., 2015; Li et al., 2017), differences in process complexity and in the degree of implementation of LULUCF practices (Arneth et al., 2017; Hartung et al., 2021; Fisher and Koven, 2020), inconsistencies in common terminology and definitions (Pongratz et al., 2014; Gasser and Ciais, 2013), and from different model assumptions and setups (Obermeier et al., 2021; Bastos et al., 2021a). Most of the investigated modelling approaches use the spatially explicit LUH2-GCB2022 dataset as LULUCF forcing (Friedlingstein et al., 2022b). The BK model H&N22
uses FAO activity data at country-level, and OSCAR22 uses both, LUH2-GCB2022 as well as FAO activity data. To analyse the impact of different land use forcing data, we further use BLUE data from the GCB2019 (hereafter, BLUE19). As the BLUE model code was not changed between the GCB2019 and GCB2022, this allows isolating the direct impact of the LULUCF forcing data, which for these GCBs was based on HYDE3.2 and HYDE3.3, respectively.

BKs do not explicitly represent biogeochemical processes and do not directly use environmental forcing data, and thus
usually exclude effects from climate change and meteorological or climate variability. DGVMs, in contrast, incorporate bio-geochemical processes implemented in their modelling scheme and – additionally to land-use change data – they use environmental forcing data as input. Consequently, DGVM estimates under transient environmental conditions include the long-term response to changing environmental conditions (e.g. atmospheric $CO_2$ increase, nitrogen deposition and fertiliser applications) and effects of climate variability. Due to the setup to calculate LULUCF emissions from DGVMs, transient DGVM $f_{\text{LULUCF}}$
estimates include the loss of additional sink capacity (LASC), representing carbon fluxes in response to environmental changes on managed land (typically croplands with low carbon sink capacity and fast turnover rates) as compared to potential natural vegetation (typically forests with large carbon sinks and low turnover rates), which leads to higher flux estimates compared to BKs towards the end of the simulated periods (Gasser and Ciais, 2013; Pongratz et al., 2014; Obermeier et al., 2021).

In contrast to modelling approaches that provide globally consistent historical $f_{\text{LULUCF}}$ analysis over the entire simulated
period (e.g. from 1700 onward), country reports cover only most recent decades for which observations and statistics exist (starting around 1990 – 2000, depending on the dataset and country), and are restricted to the territories and components that countries report. In addition, $f_{\text{LULUCF}}$ estimates from country reports rely on different definitions and assumptions than global models. One major difference is the definition of managed land, with NGHGIs having comparatively larger areas of managed land than models (Grassi et al., 2023). Further, NGHGIs estimates that rely on direct observations (e.g. national forest
inventories) include both direct and most of indirect anthropogenic effects on managed lands, which can only be separated based on modelling approaches (Grassi et al., 2018; Schwingshackl et al., 2022). Consequently, most NGHGI $f_{\text{LULUCF}}$ estimates include large parts of the natural response to recent environmental change in managed lands (e.g. larger vegetation carbon sink



due to so-called $CO_2$ fertilization), and thus, estimate larger anthropogenic $CO_2$ removals (and widely lower net $f_{LULUCF}$) than global models (IPCC, 2006; Grassi et al., 2018; Schwingshackl et al., 2022). In order to make the NGHGI and BK land use

flux estimates comparable we here translate NGHGI estimates by removing the fluxes that models attribute to the natural land sink (compare the adjusted NGHGI DB data).

Reported $f_{LULUCF}$ estimates remain highly uncertain, particularly in many developing countries, as country reports to UNFCCC do not explicitly separate managed from unmanaged forest land or report only forest-related fluxes (Grassi et al., 2022). Other countries report fluxes from additional LULUCF practices, such as from natural peatland that is converted to agriculture

or from land that is converted to human settlements (not included in modelled estimates). While human-induced degradation from logging and fires are often included in national reports, forest degradation fluxes are hardly existent in BKs.

The main differences between FAOSTAT data and the NGHGI database are explained by a more complete coverage in the NGHGI database than FAOSTAT, in particular the inclusion of non-biomass carbon pools and non-forest land uses, (Grassi et al., 2022). Additional differences stem from (1) differences in activity data, for instance different time series of forest area

which may be communicated independently to UNFCCC and FAO by different national agencies; (2) differences in emission factors, including carbon stock data and their sub-national resolution as well as their changes over time; and (3) differences in scale, especially considering that forest fluxes may be computed at grid cell or sub-national scale in many countries using remote sensing or information from national forest inventories. Additionally, the FAO area data do not distinguish between managed and unmanaged areas and thus, do not in principle separate anthropogenic and non-anthropogenic drivers (Tubiello

et al., 2021b; Grassi et al., 2022). The nature of data reported to FAO indicates that in most cases the area considered in FAOSTAT estimates is similar to the one in the NGHGIs, but often the effects of environmental changes (i.e. natural and indirect anthropogenic effects) are not included as in the NGHGIs. For this reason, we did not correct FAOSTAT estimates data here.

## 3 Data processing for country-level and regional analysis

This study compiles $f_{LULUCF}$ estimates from various modelling and country report-based approaches, aggregated to the country- and regional-level. Country-level aggregation is provided for all 186 (out of 195) UNFCCC country parties that reported LULUCF fluxes, comprising $> 99.6\%$ of global net $f_{LULUCF}$ (as derived by the three BKs). Regional aggregation is based on the 18 land regions defined by the REgional Carbon Cycle Assessment and Processes Phase 2 (RECCAP2; part of European Space Agency Climate Change Initiative (ESA CCI); Ciais et al., 2022), as defined in Tian et al. (2019) and shown in Figure A1.

Due to different spatial resolutions of the investigated datasets, preliminary processing steps were needed to obtain $f_{LULUCF}$ at country-level. Data from NGHGI DB, FAOSTAT, H&N22, and OSCAR22 are disseminated at country-level already. For the NGHGI DB and FAOSTAT estimates, we converted $CO_2$ fluxes into carbon fluxes based on their molar mass fraction, i.e. dividing by $44/12$ (according to the IPCC (2006) approach). The gridded outputs of the DGVMs and of the BLUE model were aggregated to country-level $f_{LULUCF}$ estimates based on a (modified) $0.25°$ country mask from for International Earth Science

Information Network (CIESIN). We remapped the mask of each country to the native grid of each DGVM by conservative





remapping using CDO, which yields the area fraction of every country in each DGVM grid cell. We calculate each country's $f_{\mathrm{LULUCF}}$ share by multiplying the total $f_{\mathrm{LULUCF}}$ in a grid cell with the share of each country on the total land fraction in that grid cell. The country-wide $f_{\mathrm{LULUCF}}$ is then obtained as sum over all grid cells.

Similar to the country-level aggregation, the RECCAP2 region mask was remapped to the native grid of each DGVM and the BLUE model to obtain regional aggregation. Country-level data from FAOSTAT, H&N22, NGHGI DB, and OSCAR22 was aggregated to RECCAP2 regions by overlaying the country map from for International Earth Science Information Network (CIESIN). Where country borders did not match RECCAP2 regions, namely where a country is split between two regions, those country's fluxes in FAOSTAT, H&N22, NGHGI DB, and OSCAR22 were attributed to the RECCAP2 containing the largest area fraction of the country.

## 225 4 Country-level $f_{\mathrm{LULUCF}}$ estimates from different approaches

In the main manuscript, we focus on the top eight countries with highest emissions from LULUCF as derived by the BKs in the period 1950 – 2021, and the USA. In the following, we analyse and compare $f_{\mathrm{LULUCF}}$ estimates from BKs, DGVMs, and country report-based approaches to assess the robustness of $f_{\mathrm{LULUCF}}$ estimates within and across the different datasets. Specifically, we identify countries in which the different estimates agree well and those where uncertainties in $f_{\mathrm{LULUCF}}$ estimates are

high. We further quantify and assess the gross fluxes (i.e. gross emissions and gross removals) in each country and analyse their importance in comparison to net fluxes. In the main manuscript we exemplarily focus on the eight countries with highest cumulative net LULUCF emissions since 1950 based on BKs, as well as the USA with the highest cumulative removals in this period.

Appendix A contains a comprehensive compilation of the $f_{\mathrm{LULUCF}}$ estimates from BKs, DGVMs, and country report-

based approaches for the RECCAP2 regions (Figs. A4 – A8), and for the investigated 186 countries (Figs. A9 – A18). Summary statistics of the BK estimates for all 186 country aggregates and the EU27 + UK can be found in Tables A1 – A3, and a dataset covering all aggregated country data for the period 1950 – 2021 for all datasets used is accessible under https://doi.org/10.5281/zenodo.8144174 (Obermeier et al., 2023).

### 4.1 Overview of selected country-level net $f_{\mathrm{LULUCF}}$ estimates from bookkeeping models

Country-level net $f_{\mathrm{LULUCF}}$ estimates from bookkeeping models strongly vary across countries. Net LULUCF emissions are highest (in descending order) in Brazil, Canada, China, DR Congo, India, Indonesia, Nigeria, and Russia based on cumulative estimates between 1950 and 2021 (Fig. 1a). The eight countries with the highest cumulative net emissions from LULUCF are either located in carbon-rich, forested tropical regions (Brazil, DR Congo, Indonesia, and Nigeria) and/or encompass vast territories (Brazil, Canada, China, India, and Russia). These eight top emitters emitted more than ∼53% of the total net

LULUCF emissions in the period 1950 – 2021, and are thus of outstanding importance for climate change mitigation via LULUCF emission reductions. In particular in Indonesia, the estimated emissions per area are higher than in all other 185 countries studied, which illustrates an enormous pressure on the terrestrial carbon stocks in the tropics (Fig. A3).



**Figure 1.** Net carbon fluxes from land use, land-use change and forestry ($f_{LULUCF}$) from three bookkeeping models (BKs, data from GCB2022 simulations). (a) Cumulative carbon fluxes over 1950 – 2021 and (b) average carbon fluxes in 2011 – 2021. The bars show the mean of the three BKs (filled bars), and minimum and maximum estimates (hatched bars). Numbers in parantheses show the multi-model average and standard deviation (in GtC in (a) and MtC yr$^{-1}$ in (b)). Colors indicate the absolute quantities, showing countries with net emissions in red and countries with net removals in green. All 186 country aggregates from this study are shown in decreasing order of their (a) cumulative and (b) most recent annual $f_{LULUCF}$. In each panel, the top ten emitters and the five countries with the largest removals are labelled (and the countries from the main manuscript if not yet included). The dashed red lines show the percentiles of net carbon emissions for each panel when adding the countries in decreasing order.

During the second half of the 20th century, high net LULUCF hotspots became increasingly concentrated in the countries of the Global South. As stated in the GCB2022 (Friedlingstein et al., 2022b), more than 50% of most recent net emissions

from LULUCF occur in only three countries, namely Brazil, DR Congo, and Indonesia – all located in the tropics (Fig. 1b). The pronounced land-use change impact in the tropics are also evident from the $f_{LULUCF}$ estimates calculated per area and per capita, with the ten largest net emitters all located in the tropics (Fig. A3 & A2). However, most of these emissions are





embodied in trade and are caused by consumption in industrialized regions such as Europe, the United States, and China (Hong et al., 2022). A trend towards fewer countries comprising larger shares of global net LULUCF emissions is found when

comparing cumulative and most recent annual $f_{\mathrm{LULUCF}}$ estimates. In the period 1950 – 2021, the top 22 net emitters comprised more than 75% (top 43 emitters more than 90%) of cumulative net emissions from LULUCF, while in 2011 – 2021, only 15 countries make up for 75% (top 33 emitters more than 90%) of the global net emissions. In China, the $f_{\mathrm{LULUCF}}$ estimates from the BKs show an remarkable turnaround, turning China from the third highest cumulative emitter in 1950 – 2021 to the country with the third highest net removals in 2011 – 2021.

**Table 1.** Statistics of the annually averaged (2011 – 2021) and cumulative (1950 – 2021) net carbon fluxes from land use, land-use change and forestry ($f_{\mathrm{LULUCF}}$) for the nine countries analysed. The table indicates the mean $f_{\mathrm{LULUCF}}$ estimates, its standard deviation (SD), and its relative uncertainty (SD divided by the absolute mean value).

| Country | Cumulative $f_{\mathrm{LULUCF}}$ in 1950 – 2021 (GtC) | | | Annual mean $f_{\mathrm{LULUCF}}$ in 2011 – 2021 (MtC yr$^{-1}$) | | |
| --- | --- | --- | --- | --- | --- | --- |
| | Mean | SD | Rel. Unc. (%) | Mean | SD | Rel. Unc. (%) |
| Brazil | 21.8 | 7.0 | 32 | 285.3 | 111.6 | 39 |
| Indonesia | 14.0 | 1.1 | 8 | 283.1 | 16.3 | 6 |
| China | 4.8 | 7.3 | 150 | −9.0 | 62.9 | 700 |
| Congo, Dem. Rep. | 4.6 | 0.6 | 13 | 155.3 | 21.8 | 14 |
| India | 3.3 | 2.3 | 71 | 15.3 | 26.5 | 170 |
| Canada | 2.9 | 1.7 | 57 | 23.2 | 8.2 | 35 |
| Russian Federation | 2.3 | 3.1 | 140 | 6.3 | 63.9 | 1000 |
| Nigeria | 2.2 | 0.9 | 39 | 6.8 | 4.8 | 71 |
| United States | −1.0 | 7.0 | 680 | −26.7 | 57.3 | 210 |

The large number of net emitting countries and the fact that BKs estimate a global net source of carbon from LULUCF to the atmosphere is in stark contrast to the pledges to achieve the goals associated with the Paris Agreement and the reported global carbon removal from LULUCF when summed across all NGHGIs (Grassi et al., 2022). Of note, despite widely included net negative emissions from LULUCF in the NDCs, including the need for carbon dioxide removal (CDR) techniques, BKs estimate net removals from LULUCF only for very few countries (both cumulatively between 1950 – 2021 as well as more

recently in 2011 – 2021). Substantial country-level net removals from LULUCF are only found in the USA, some European countries, and, in the most recent decade, in China. However, large relative uncertainties in BK estimates remain, within the cumulative net $f_{\mathrm{LULUCF}}$ estimates (globally ∼38%, and much higher in specific countries, e.g., in China ∼150%, USA ∼680%; see Tab. 1) and recent annual $f_{\mathrm{LULUCF}}$ estimates (globally ∼31%, in China ∼700% and Russia ∼1000%). In order to explain the uncertainties related to $f_{\mathrm{LULUCF}}$ estimates from BKs, the latter need to be compared to estimates from other approaches and

the underlying drivers for the differences in $f_{\mathrm{LULUCF}}$ estimates need to be investigated.

### 4.2 Comparing country-level net $f_{\mathrm{LULUCF}}$ estimates from BKs and DGVMs

To test the robustness and explain the large spread in modelled country-level $f_{\mathrm{LULUCF}}$ estimates, we compare and discuss the differences of the net estimates from the BK ensemble and the DGVM ensemble with respect to the characteristics of the

individual modelling approaches, particularly regarding the underlying land use forcing data and, for the DGVM ensemble,

the environmental forcing data.

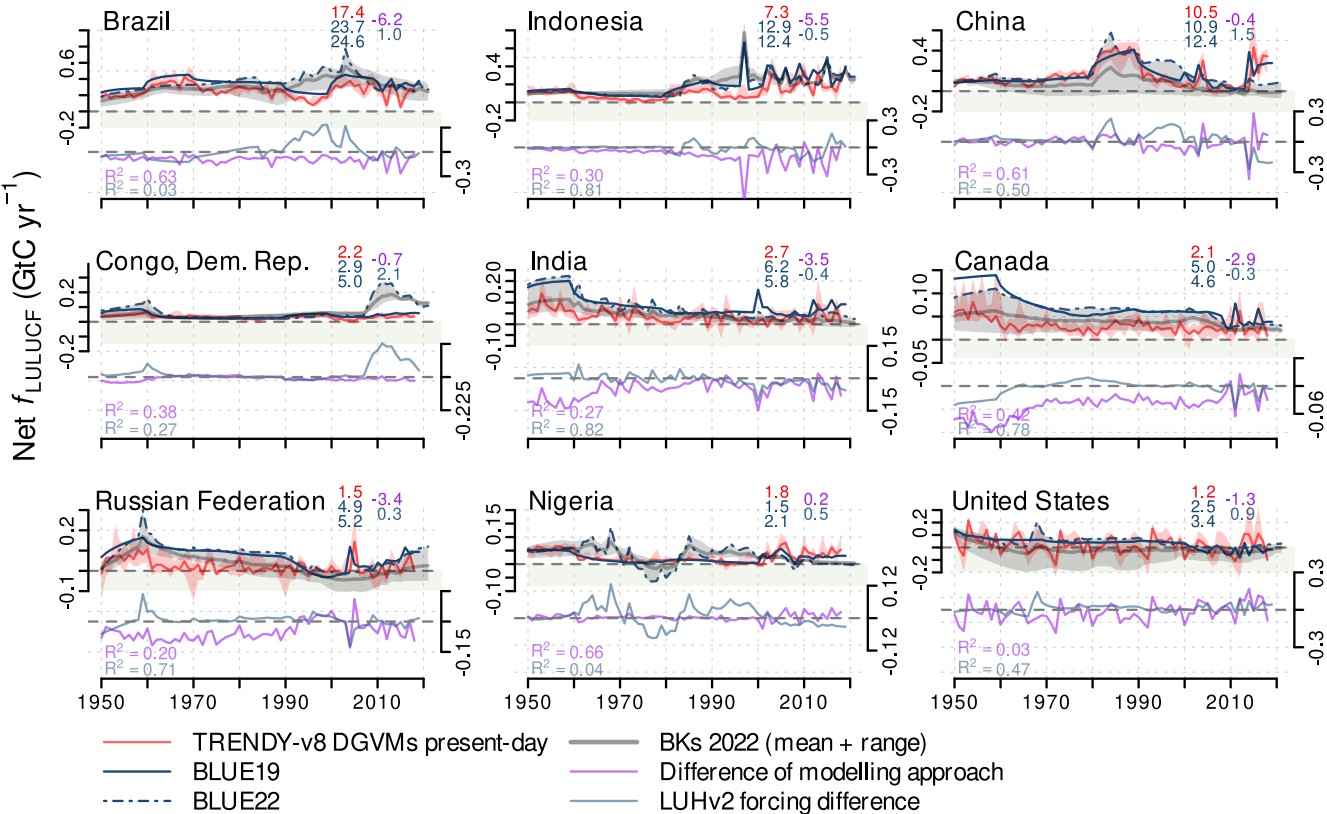

**Figure 2.** Time series of net carbon flux from land use, land-use change and forestry ($f_{LULUCF}$) in 1950 – 2021 derived by bookkeeping models (BKs) and TRENDYv8 simulations with dynamic global vegetation models (DGVMs; used in the GCB2019) under present-day climate forcing. The eight countries with highest cumulative emissions since 1950 are shown in decreasing order, plus the USA. The figure shows the mean and absolute range of three BKs (using the GCB2022 simulations, BKs 2022) and the median and interquartile range of the eight DGVMs. Additionally, estimates from BLUE simulations from the GCB2019 (BLUE19; blue solid) and from the GCB2022 (BLUE22; blue dashed) are shown to illustrate the impact of updates in the LUHv2 forcing data (difference shows in steelblue), and the differences between DGVMs and BLUE (TRENDYv8 present-day minus BLUE19; purple) to illustrate the relevance of different modelling approaches. Numbers in the top right corner of each panel indicate the median of the cumulative $f_{LULUCF}$ sums over 1950 – 2018 (in GtC) for TRENDYv8 (red), BLUE19 (blue), BLUE22 (blue), the difference TRENDYv8 present-day minus BLUE19 (purple) and of the LUHv2 forcing (BLUE22 minus BLUE19; steelblue); numbers in the bottom left corner indicate coefficients of correlation squared for BLUE19 and TRENDYv8 (purple), and for BLUE19 and BLUE22 (steelblue). BLUE19 data is only available until 2019 and TRENDYv8 data are only available until 2018. Greenish background depicts negative $f_{LULUCF}$, that is carbon removal from the atmosphere.



Country-level net $f_{\text{LULUCF}}$ estimates of BKs and DGVMs from 1950 onward agree generally well in the investigated countries (Fig. 2; refer to Fig. A4 for RECCAP2 regions and Figs. A9 and A10 for all 186 investigated countries).

In most countries, modelled estimates from BLUE19 and (present-day) DGVM simulations show consistent temporal evolutions and peaks in emissions, when using the same land use forcing data (compare the "difference of the modelling approach",
derived as the present-day TRENDYv8 mean minus BLUE19 which share the same (LUHv2) land use forcing data; purple line in Fig. 2). Rather low coefficients of correlation for the estimates from the different modelling approaches partly result from the inter-annual variability that is captured in the DGVMs but usually not in the BKs. Yet, the $f_{\text{LULUCF}}$ estimates substantially differ in some of the countries despite identical land use forcing data. The most striking difference occurs in 1997 in Indonesia. 1997 was an El Niño year causing high carbon emissions from organic soils in Indonesia, which are included in the BK
estimates but not in the used DGVM estimates (in line with the GCP assessments).

BLUE19 generally yields higher net emission estimates compared to the present-day TRENDYv8 ensemble (except for Nigeria), which indicates a tendency towards higher estimates when using the BK approach. To investigate this further, we compare the BLUE22 estimates with the 2022 estimates of all three BKs (BKs 2022; note that BLUE model code was not changed between 2019 and 2022 versions). BLUE22 tends to estimate the highest emissions among the BKs in most coun-
tries and during most of the time, indicating that the BK mean might agree better with the mean of the present-day DGVM simulations compared to BLUE19. This can mainly be explained by high carbon densities assumed in the BLUE model but also due to BLUE capturing the full extent of (LUH2-based) gross transitions (compare Sect. 4.3 and description of the BKs in Sect. A1).

The strong influence of the land use forcing data is highlighted by the often differing trends and peaks in $f_{\text{LULUCF}}$ from
BLUE19 based on HYDE3.2 and BLUE22 based on HYDE3.3 (compare the "LUHv2 forcing difference", derived as BLUE22 minus BLUE19, in Fig. 2) and the huge ranges in the BK estimates in many regions. LUHv2 forcing differences are largest in Brazil, China, DR Congo, and Nigeria where the BLUE model estimates are considerably larger when using the more recent HYDE3.3 data. In contrast, BLUE22 has lower $f_{\text{LULUCF}}$ estimates than BLUE19 in Canada (mainly in the 1950s), China, India, Indonesia and Nigeria (for the latter four, mainly in the most recent decades). In particular for tropical countries, lower
emissions may be related to decreased cropland expansion in the updated HYDE data (Friedlingstein et al., 2022a).

Emission peaks around 1960 occur in several regions (e.g. Canada, DR Congo, and Russia) mainly in the 2022 estimates (BLUE22) but not in the 2019 estimates (BLUE19). Those peaks can be explained by an artifact in HYDE3.3, resulting from the merge of historical HYDE data (used up to 1960) with FAOSTAT data (used from 1961 onward) (Chini et al., 2021; Friedlingstein et al., 2022a). The 1960 peaks thus do not represent any real-world land-use change fluxes, and should be
corrected in future updates of HYDE (as has already been achieved for Brazil, compare Sect. A1 and Friedlingstein et al., 2022a; Rosan et al., 2021). For Brazil, an additional peak in 2004 corresponds to increased deforestation for cropland and pasture, followed by a slowdown in deforestation rates due to governmental regulations, which is only captured in HYDE3.3 through the inclusion of ESA CCI Land Cover data (Rosan et al., 2021).

Additionally, the environmental forcing plays an important role in many regions, as can be assessed by comparing transient
and present-day TRENDYv8 simulations (Fig. 3 – the "Present-day versus transient difference"; compare low values for the

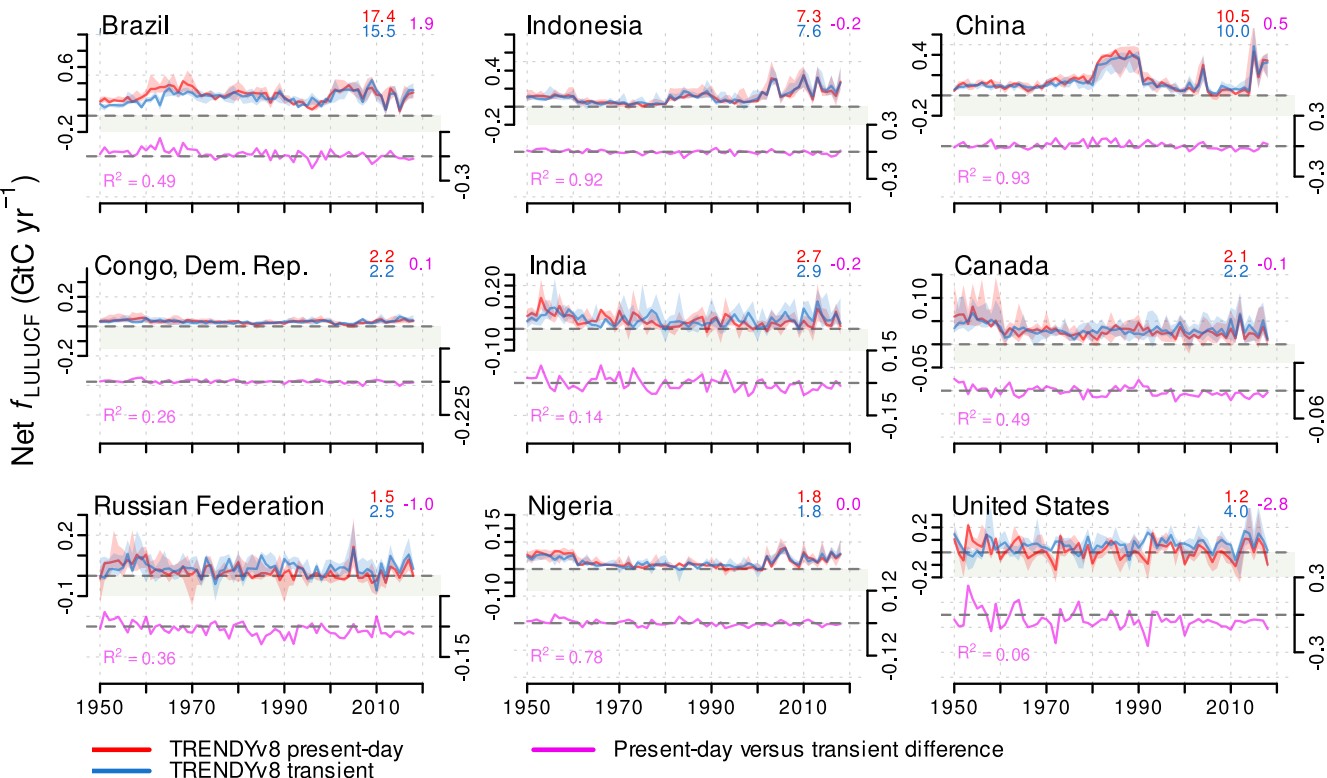

**Figure 3.** Net carbon flux from land use, land-use change and forestry ($f_{LULUCF}$) in 1950 – 2021 derived from TRENDYv8 simulations with dynamic global vegetation models (DGVMs) under historical (transient) and fixed present-day environmental conditions (compare appendix). The eight countries with the highest cumulative emissions since 1950 are shown in decreasing order, and the USA. The median and interquartile range of eight DGVMs are shown. Lines (shading) indicates the median (interquartile range) of the eight DGVMs, for which data for both simulations are available. Lines in the lower panels indicate the impact of the environmental forcing (present-day minus transient; purple). Numbers in the top right corner depict the multi-model median of the cumulative sums in the period 1950 – 2019 (left column; GtC) and the differences between the simulations (right column); numbers in the bottom left corner indicate the coefficient of correlation squared for TRENDYv8 present-day and transient simulations (purple). The light green background indicates negative $f_{LULUCF}$, that is net carbon removal from the atmosphere by LULUCF.

coefficients; refer to Fig. A5 for RECCAP2 regions and Figs. A11 and A12 for all 186 investigated countries). Estimates of $f_{LULUCF}$ under present-day environmental forcing tend to be higher compared to transient estimates in the earliest simulated periods, particularly in tropical regions with high LULUCF activity (e.g., in Brazil, China, DR Congo, and India), which is also reflected in the cumulative emissions estimates (see numbers displayed in Fig. 3). This can be explained by higher carbon stocks

under present-day environmental forcing compared to transient forcing (the multi-DGVM mean global vegetation carbon stock increased by ∼23% from 664 to 815 PgC from 1800 until today), particularly in early simulation periods (when atmospheric $CO_2$ concentration was substantially lower than today; Obermeier et al., 2021). Differences between present-day and transient





$f_{\text{LULUCF}}$ estimates become smaller as the simulations progress, as transient environmental conditions approach present-day conditions and additionally accumulate the loss of additional sink capacity (Pongratz et al., 2014). Noteworthy, towards the end

of the simulated period, transient $f_{\text{LULUCF}}$ estimates even tend to exceed present-day estimates, due to the steadily accumulating loss of additional sink capacity, which globally comprises ∼0.8±0.3 GtC yr$^{-1}$ (∼40%) of transient $f_{\text{LULUCF}}$ in the period from 2009 – 2018 (Obermeier et al., 2021). In the temperate zone (e.g., in Russia and the USA), variations in $f_{\text{LULUCF}}$ estimates due to environmental conditions rather depend on the inter-annual meteorological and climate variability. Here, carbon stocks have not been enhanced by higher $CO_2$ concentrations as homogeneously as in the tropics and climate change has even led to

decreased carbon stocks in some regions (Obermeier et al., 2021).

As described above (and in the appendix), the multiple modelling approaches differ in their underlying assumptions, their implemented process complexity and parametrizations, and the used input forcing data. These differences partly lead to highly differing estimates, in particular at finer spatial scales, which decreases the accuracy of some country-level estimates from models. The differences in country-level net $f_{\text{LULUCF}}$ estimates that are due to the modelling approach versus changes in land

use forcing (approximated by the LUHv2 forcing difference for the BK model BLUE) or environmental forcing (difference between present-day and transient DGVM simulations) are of similar order of magnitude, yet with very different spatial patterns. Whether the modelling approach, land use or environmental forcing is the dominating factor depends on the specific country and partly also on the specific time period. The modelling approach had the highest influence on the cumulative estimates, for example, in Brazil, Canada, India, Indonesia, and Russia, whereas the LUHv2 forcing difference in the BK model BLUE was

higher in China, DR Congo and Nigeria. For the DGVMs, the land use forcing data impacted cumulative $f_{\text{LULUCF}}$ estimates widely more strongly compared to the environmental forcing (except in the USA), although of similar magnitude.

### 4.3 Net $f_{\text{LULUCF}}$ from individual bookkeeping models at country-level and country report-based estimates

Despite broad agreement in most country-level net $f_{\text{LULUCF}}$ estimates when averaged across multiple models, individual BK model output differs strongly in some countries (compare BK range in Figs. 2 and 4) and, not surprisingly, country report-

based estimates mostly diverge even more (Fig. 4; refer to Fig. A6 for RECCAP2 regions and Figs. A13 and A14 for all 186 investigated countries).

Differences in annual net $f_{\text{LULUCF}}$ estimates across the three BKs (based on simulations with the most recent land use forcing) are particularly high in Canada and the USA throughout the 20th century, and before and after emission peaks (e.g., 1960 and 2011 in DR Congo, 1980s in China, and in the 1950s in India). As stated above, the high differences related to emission peaks

are predominantly due to the use of different land use forcing data, which is reflected by the fact that H&N22 (based on FAO data) does not capture any of theses peaks while BLUE22 and OSCAR22 models (both of which using LUH2 data) show very similar trends and peaks. The estimates of OSCAR22 often lie close to the multi-model mean of the BKs, which can be explained by the fact that OSCAR22 uses both FAO and LUH2 forcing data to derive a best-guess estimate for $f_{\text{LULUCF}}$ (Gasser et al., 2020). The huge uncertainties in Canada, China, and the USA cannot directly be explained by the net $f_{\text{LULUCF}}$ but are

discussed in Sect. 4.4 in relation to the gross fluxes.



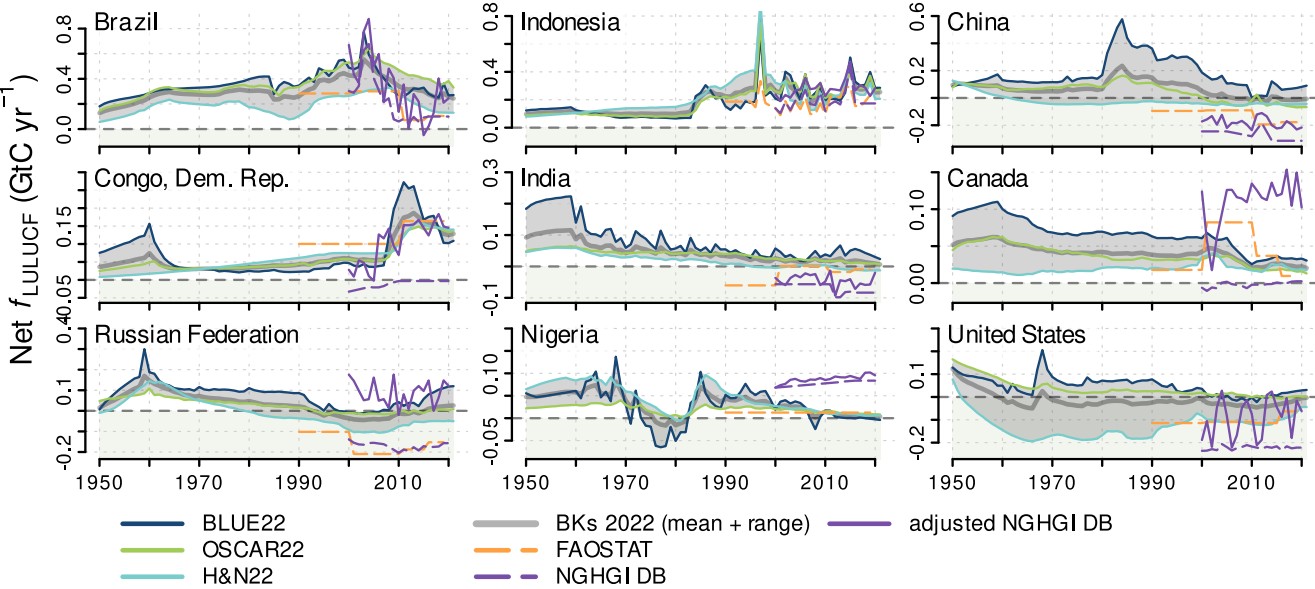

**Figure 4.** Net carbon flux from land use, land-use change and forestry ($f_{\mathrm{LULUCF}}$) in 1950 – 2019 derived from individual bookkeeping models (BKs) used in GCB2022 (BKs 2022) and country report-based estimates (FAOSTAT and NGHGI DB). The use of dashed and solid lines indicates that BK estimates and country report-based estimates are not directly comparable (see Sect. 2.5 and the appendix). For better comparability, the adjusted NGHGI DB estimates, matching the $f_{\mathrm{LULUCF}}$ definition of the BKs, are additionally shown. The gray line (shading) depicts the median (range) of the three BKs. The light green background indicates negative $f_{\mathrm{LULUCF}}$ that is net carbon removal from the atmosphere by LULUCF.

Beyond the effects of the land use forcing data, BK model differences in net $f_{\mathrm{LULUCF}}$ can be explained by several individual model specifics. The upper limit of the BK range is predominately defined by BLUE22, particularly during phases of high BK uncertainty, while the lower limit is often defined by H&N22. This is mainly due to different assumed carbon densities, which are particularly high in the BLUE model and low in the H&N model (compare appendix and Bastos et al., 2021a, b).

Moreover, the inclusion of sub-grid scale transitions in BLUE22 (and OSCAR22) increases emission estimates compared to H&N22. Additionally, H&N22 assumes conversion of natural grasslands to pasture, while BLUE22 and OSCAR22 allocate pasture proportionally on all natural vegetation that exists in a grid-cell (Hansis et al., 2015; Friedlingstein et al., 2022a), yielding lower $f_{\mathrm{LULUCF}}$ for H&N22. In addition, different turnover periods for the HWPs in the different BKs lead to varying BK estimates after significant changes in LULUCF practices. The exception of slightly higher emission estimates from H&N22

in Indonesia (mainly in the 1990s) and DR Congo (from 1970s until 2007) are due to much higher carbon removals due to afforestation and reforestation in BLUE22 and OSCAR22 (not shown) and faster increasing emissions from deforestation in the H&N22 model.

Country report-based net $f_{\mathrm{LULUCF}}$ estimates are substantially lower than BK estimates in most of the investigated countries, in particular NGHGI DB has the lowest emission estimates in almost all investigated countries. Much of this discrepancy




(globally adding up to ~1.6 GtC yr$^{-1}$ for the period 2001 – 2020) can be explained by different definitions, in particular, by the inclusion of natural and indirect human-induced fluxes on a broader area of managed land than BKs, such as those resulting from increased forest regrowth due to higher atmospheric $CO_2$ concentration and N deposition, in many country reports (compare appendix and Grassi et al., 2023; Schwingshackl et al., 2022). This is confirmed by the fact that the adjusted NGHGI DB data, where the natural land sink on managed land is subtracted, agree better with the BK estimates than the
NGHGI DB data for most countries.

However, for individual countries, other reasons, such as omitted fluxes due to incomplete reporting of land uses and carbon pools, also (partly) explain the differences between modelled and reported $f_{LULUCF}$ estimates (Schwingshackl et al., 2022). In the USA, lower country report-based estimates may additionally result from the inclusion of $CO_2$ removals from urban vegetation (Churkina et al., 2010), which is not considered in the BK estimates. In addition, despite of the increased methodological
complexity of the approaches used by many developed countries, it was shown, that reported $f_{LULUCF}$ estimates for the USA still remain uncertain, mainly due to uncertainty in inputs, model parameters, and plot-based sampling (e.g. McGlynn et al., 2022). The Canadian NGHGI report discounts fluxes (emissions and removals) on areas affected by wildfires and severe insect disturbances and reports them in a separate category (Kurz et al., 2018) which can explain the strongly increased difference for the adjusted NGHGI DB estimate (Grassi et al., 2018; Schwingshackl et al., 2022).In China, the large gap between country
report-based and modelled estimates might be explained by high carbon removals from afforestation and ecological restoration projects (Yang et al., 2022; Jin et al., 2020; Yu et al., 2022) considered in the country reports but not fully included in the land use data used by the models (compare Sect. and Yu et al., 2022).

The generally good agreement in the $f_{LULUCF}$ estimates from BKs and inventories in Indonesia is due to the dominance of the added peat data, with a pronounced interannual variability controlled mostly by El Niño–Southern Oscillation patterns on
top of land use (Federici et al., 2017).

LULUCF flux estimates from FAOSTAT are largely in agreement with BK estimates, with some important differences, notably lower estimates in China and Russia and higher estimates in DR Congo and Canada. FAOSTAT estimates are generally higher than the NGHGI DB estimates except for the period from 2000 – 2019 in Russia and Nigeria and the emission peak in Brazil in 2004. As for the NGHGI DB estimates, lower FAOSTAT estimates compared to BK estimates are likely due to the
inclusion of all fluxes on managed land (compare adjusted NGHGI DB data). Differences between FAOSTAT and NGHGI DB estimates can be explained by the generally more complete coverage of carbon fluxes in the latter and differing approaches to estimate forest fluxes, where FAO applies a carbon stock change approach based on observed forest data from FRA, while NGHGI reports are based on the use of a simple carbon stock change approach or a gain loss approach by scaling up of forest growth rates based on IPCC default factors to forest land estimates (refer to appendix and Grassi et al., 2022; Tubiello et al.,
2021b for a detailed description of the differences between UNFCCC country, NGHGI DB and FAOSTAT estimates).

Additionally, the underlying data on forest land differ, with the NGHGI DB database reporting much greater forest areas and forest carbon removals (in particular for Non-Annex I countries). Moreover, NGHGIs of Annex I and the largest Non-Annex I countries include also non-biomass carbon pools and non-forest land uses, while, except for organic soils, FAOSTAT includes above- and belowground biomass pools only (Federici et al., 2015; Tubiello et al., 2021b; Grassi et al., 2022). The





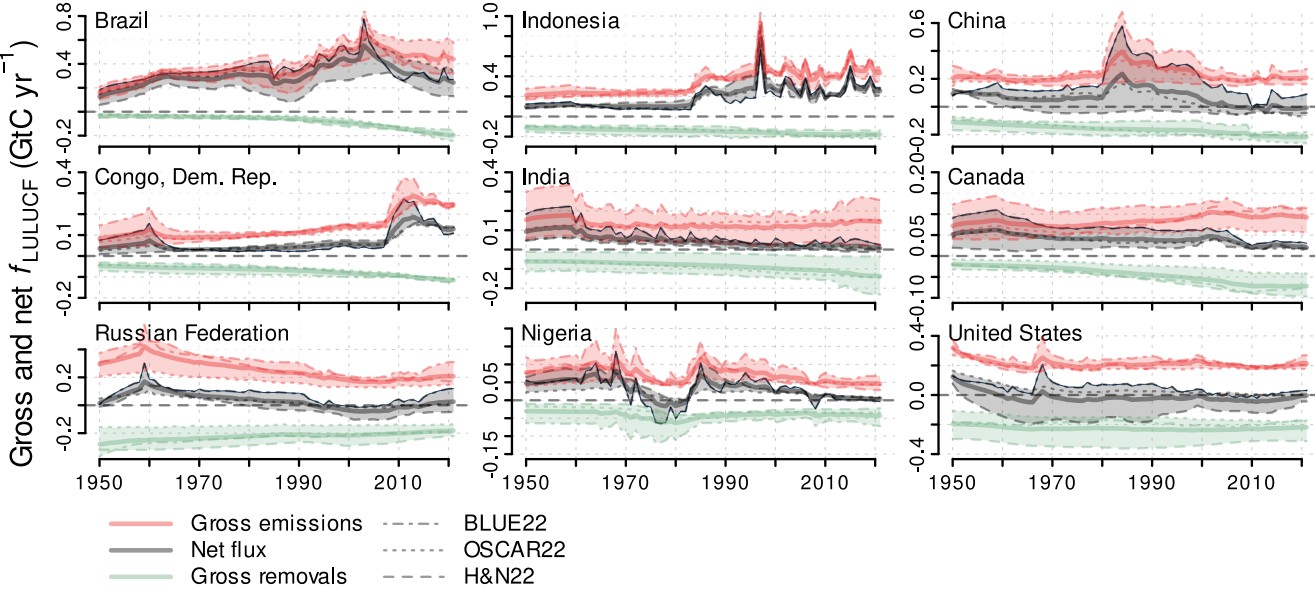

**Figure 5.** Gross and net fluxes from land use, land-use change and forestry ($f_{\text{LULUCF}}$) from 1950 until 2021 derived by three bookkeeping models (BKs; used in the GCB2022). Eight countries with the highest cumulative emissions since 1950 are shown in decreasing order, plus the USA. Lines (shaded area) depict the mean (range) of the BK estimates.

higher estimates of NGHGI DB compared to FAOSTAT in Brazil are caused by larger deforestation and afforestation areas in the Brazilian report to UNFCCC compared to FRA, and the fact, that it considers gross deforestation and afforestation while FAOSTAT reports net deforestation and afforestation directly (Federici et al., 2017; Rosan et al., 2021; Schwingshackl et al., 2022) (which might increase emissions in the NGHGI due to the asymmetry in instantaneously occurring gross emissions versus slowly increasing gross removals over the long-term).

### 4.4   Gross $f_{\text{LULUCF}}$ from bookkeeping models at country-level

To get more insights into the underlying drivers of country-level net LULUCF estimates, we split them into gross fluxes, namely gross emissions (or "sources") and gross removals (or "sinks"). As stated in the Introduction, we here define these gross fluxes as the sum of all fluxes related to those LULUCF practices that typically lead to emissions or removals, respectively. Gross emissions are mainly caused by deforestation, peatland degradation, biomass burning, the decay of HWPs and biomass left

on site after harvest (Friedlingstein et al., 2022b). Gross removals are mainly associated with afforestation and reforestation including forest regrowth after agricultural abandonment, as well as forestry cycles and the restoration of other (non-forest) ecosystems. In the following, we present and discuss gross $f_{\text{LULUCF}}$ derived from the three BKs for the nine selected countries (Fig. 5; refer to Fig. A7 for RECCAP2 regions and Figs. A15 and A16 for all 186 investigated countries).



Similarly as for the net fluxes, gross fluxes modelled by the three BKs show widely similar trends and agree on the timing
of emission peaks. Peaks in net emissions are predominantly due to peaks in gross emissions while the time-series of gross
removals is much smoother, consistent with the slower pace of vegetation regrowth. Consequently, the short-term evolution of
net $f_{\mathrm{LULUCF}}$ is much more influenced by the dynamics of highly fluctuating gross emissions than by the dynamics of rather
slowly changing gross removals. However, the decreasing trends in net $f_{\mathrm{LULUCF}}$ estimates across many regions (globally from
$1.6 \pm 0.7$ GtC yr$^{-1}$ in the 1960s to $1.1 \pm 0.7$ GtC yr$^{-1}$ in the period from 2011 – 2020) mainly relate to a steady increase in the
gross removals (globally from $-1.9 \pm 0.4$ GtC yr$^{-1}$ to $-2.7 \pm 0.4$ GtC yr$^{-1}$ in the same period), which exceeded the increase
in gross emissions (globally from $3.4 \pm 0.9$ GtC yr$^{-1}$ to $3.8 \pm 0.6$ GtC yr$^{-1}$ in the same period; Friedlingstein et al., 2022a).
This smooth evolution towards increased removals results from increased carbon sequestration on previously managed land,
mainly due to forest regrowth and soil recovery. Environmental changes, such as those resulting from $CO_2$ fertilization, are not
modelled by BKs (except the changing carbon densities in OSCAR).

In most regions, uncertainties in net $f_{\mathrm{LULUCF}}$ are due to uncertainties in gross emissions rather than uncertainties in gross
removals. The largest differences, and thus pronounced uncertainties, in gross emissions from BKs are found for India and
Canada as well as in the years before and after most emission peaks in many regions. Uncertainties related to the peaks in the
1960s mainly stem from the merging of two different datasets. In China, the pronounced peak in the 1980s is caused by spurious
signals in the LUHv2 data, inherited from an abrupt cropland increment in the FAO data (Yu et al., 2022). Because cropland
area is quantified relative to forest proportions an increasing cropland area causes decreasing forest area (and vice versa), while
China's afforestation projects were largely implemented in drier and previously unmanaged and unforested lands, increasing
the total forest area without replacing croplands (Yu et al., 2022). Similar to net $f_{\mathrm{LULUCF}}$, the highest emission estimates are
generally derived from BLUE22 and the lowest emissions predominantly from H&N22. As stated above, this can mainly be
explained by different process representation and parametrization in the models (compare Sec. 4.3). The exception of higher
OSCAR22 estimates in Brazil in recent decades likely results from higher deforestation rates since 2004 and shorter turnover
times for HWPs in OSCAR compared to the other BMs. In addition, OSCAR uses the averaged biome-specific carbon densities
taken uniformly over the country which may overestimate emissions in particular in large countries covering differing types
of the same biome (e.g. different types of forest), if land use transitions predominantly happen in regions with lower carbon
densities.

The highest differences in gross removal estimates among the BKs are found in India, Russia, and the USA. In India, this
may result from greater removals due to the inclusion of sub-grid scale transitions in BLUE22 and OSCAR22, while H&N22
estimates rather negligible removals. Noteworthy, in India, the large uncertainties in gross emissions and removals from BKs
translate into a huge uncertainty in net $f_{\mathrm{LULUCF}}$ in the 1950's, but subsequently uncertainties in gross fluxes cancel out, yielding
only small uncertainties for the net flux. In Russia, the models agree in a decreasing removal trend despite a considerable
spread with large removal estimates by H&N22 and small estimates by OSCAR22. In recent decades, this decreasing removals
in Russia can partly be explained by the decreasing trend in the abandonment sink as was shown for the BLUE model by
Winkler et al. (2023) in addition to intensified logging and wood harvest activities that cause ongoing deforestation (Kuzminyh
et al., 2020). In the USA, the large BK range for net fluxes is predominantly due to large uncertainties in the removal estimates,





while the gross emission estimates agree well among BKs. This removal-driven net $f_{LULUCF}$ uncertainty can be explained by

the inclusion of fire management in the USA in H&N22 leading to large removal estimates, while BLUE22 and OSCAR22 show much lower removals estimates.

## 4.5   Ratio of net-to-gross $f_{LULUCF}$ from bookkeeping models

To further investigate the importance of gross fluxes, we calculate the ratio of net $f_{LULUCF}$ to the sum of gross $f_{LULUCF}$ (net-to-gross ratio, with the sum of gross fluxes defined as the range between gross emissions and gross removals) for the three

BKs (Fig. 6; refer to Fig. A8 for RECCAP2 regions and Figs. A17 and A18 for all 186 investigated countries). Ratios close to 1 (close to -1) indicate that the net flux reflects mostly gross emissions (gross removals) and only very small gross removals (gross emissions). Ratios between 0 and 0.5 (between 0 and -0.5) indicate that the net flux represents only a small fraction of the occurring gross sources (sinks), which is the case in most countries during most of the time investigated. Ratios close to zero indicate that gross fluxes are largely compensating each other, which might indicate a sustainable land management that

causes gross removals to largely offset gross emissions.

Seven out of the nine investigated countries (Brazil, China, India, Canada, Russia, Nigeria, and the USA) show decreasing country-level ratios of net-to-gross $f_{LULUCF}$ from 1950 onward, indicating increasingly compensating gross emissions and gross removals. This is mostly due to increasing gross removals from LULUCF, while at the same time the gross emissions did not increase in such a pronounced way (particularly in Brazil, Canada, China, India, and Nigeria). In some of the countries, the

ratios became even negative over time (China, India, Russia, and the USA), indicating that gross removals became larger than gross emissions. Large negative net-to-gross ratios indicate that gross removals are much larger than gross emissions, and thus the (negative) net flux is mostly controlled by carbon removals from the atmosphere (e.g., in most recent decade in European countries, Japan and Turkey; compare Fig. 7a) and Tables A1 – A3).

In contrast, increasing net-to-gross ratios over time are found in Indonesia and DR Congo (particularly, in the most recent

decades), mainly due to strongly increasing gross emissions, which are not compensated by equally large gross removals, despite increasing removals also observable in these countries (see Fig. 5). High positive net-to-gross ratios in the most recent decade reveal large gross emissions that are not compensated by gross removals, and are mainly found in the tropics and the Southern Hemisphere, in particular in Argentina, Angola, Paraguay, Bolivia, Papua New Guinea and Tanzania (Fig. 7a).

Large uncertainties in the net-to-gross ratio in Canada and China in the second half of 21st century are caused by strongly

varying emission estimates, and in the same period in the USA, from strongly varying removal estimates.

Near zero net-to-gross ratios indicate that gross fluxes counterbalance each other, i.e. gross removals compensate gross emissions. Country-level near zero net-to-gross ratios can be found in China, India, Russia and the USA, particularly in the most recent decades (compare Figs. 6 and 7a). Grid cell-wise analysis revealed, however, that pronounced gross fluxes occurred also in these countries. This highlights that considering only the net land use $CO_2$ fluxes might miss the importance of potentially

large gross fluxes. This is especially true when net fluxes are estimated on a larger scale, such as at the country-level and particularly for very large countries, since here the opposing gross fluxes, which often occur spatially separately, are more likely to be offset (compare, for example, USA and China in Fig. 7c and d). This, furthermore, highlights the need for spatial



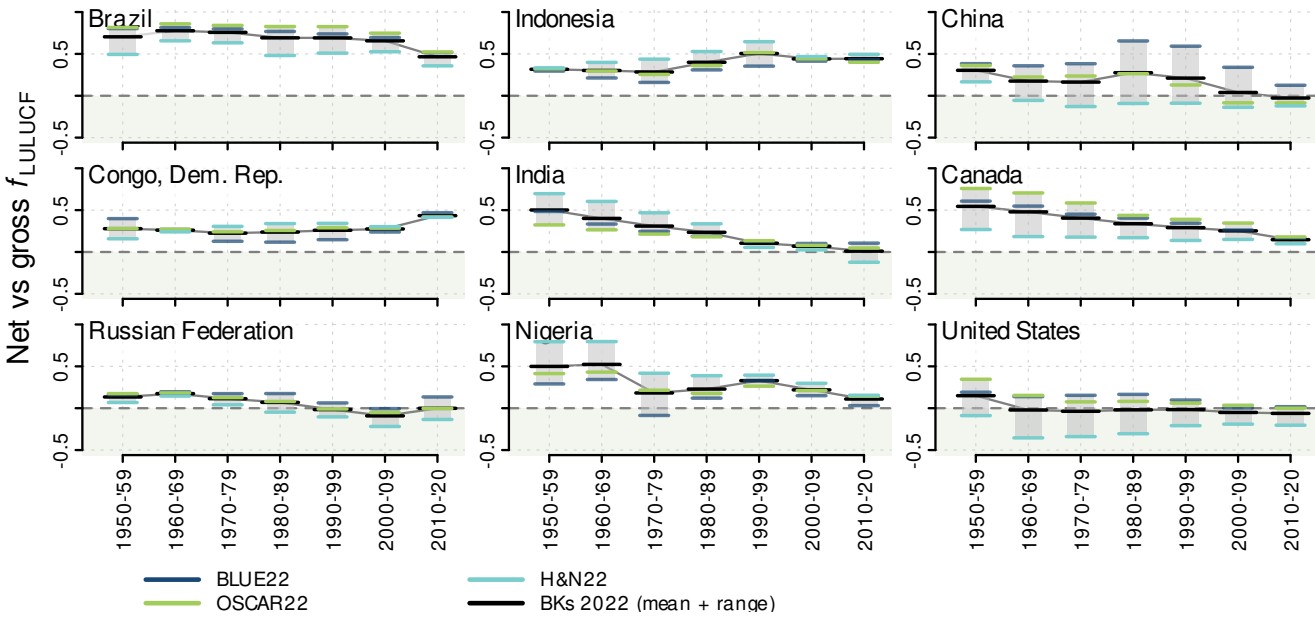

**Figure 6.** Decadal mean ratio of net-to-gross fluxes from land use, land-use change and forestry (with "gross" defined as the range between gross emissions and gross removals) from 1950 to 2020 derived by three bookkeeping models (BKs). The eight countries with the highest cumulative emissions since 1950 are shown in decreasing order, plus the USA. Values close to 1 or -1 indicate that either gross removals or gross emissions are close to 0 and the net value corresponds to either gross emissions or removals, with little compensating effects. Values near zero indicate that emissions and removals largely compensate each other. A negative ratio (green background) indicates net removals, that is, gross removals greater than gross emissions.

explicit analysis of the net as well as gross $f_{\mathrm{LULUCF}}$ and that country commitments based on net LULUCF fluxes can still be associated with large emission fluxes. Similarly, the rather vague commitment of the Glasgow Leaders' Declaration on Forests
and Land Use to halt deforestation, without stating whether these accounts for gross or net transitions, can lead to strongly varying forest flux trajectories (Gasser et al., 2022). In line with Gasser et al. (2022), we therefore argue that climate mitigation measures should focus on gross fluxes from LULUCF rather than net fluxes.

## 5 Conclusions

In this study, we have compiled country-level data on carbon fluxes from land use, land-use change and forestry ($f_{\mathrm{LULUCF}}$) from
model- and country report-based approaches comprehensively for 186 countries. The increasing spatial resolution of modelling approaches makes it possible to provide model-based $f_{\mathrm{LULUCF}}$ estimates at country-level, which can be compared to estimates based on official country reports. The comparison of multiple approaches for estimating $f_{\mathrm{LULUCF}}$ showed a fair agreement in the majority of countries, although with large differences in some other countries. The modelling approaches (BKs and DGVMs)



**Figure 7.** Maps of (a) country-level ratio of net-to-gross carbon fluxes from land use, land-use change and forestry (with "gross" defined as the range between gross emissions and gross removals), (b) net $f_{LULUCF}$, (c) gross emissions, and (d) gross removals as average of the three bookkeeping models for the period 2011 – 2021. Hatching in (a) indicates countries with a very low range in gross fluxes (average gross emissions minus gross removals smaller than 0.1 t C ha$^{-1}$ yr$^{-1}$. Green (brown) colors in (a) depict negative (positive) net fluxes with a value of -1 (+1) indicating that no carbon emissions (removals) occur. The maps depict the median from three bookkeeping models (BLUE22, H&N22, OSCAR22). Grid cells with a gross flux range smaller than 0.02 tC ha$^{-1}$ yr$^{-1}$ are excluded. Gross fluxes from OSCAR22 and H&N22 were distributed using the spatial patterns of the gross flux density in BLUE for each country respectively.

yield generally consistent $f_{LULUCF}$ estimates for the nine investigated countries. Differences, particularly across BKs, are due

to differences in land use forcing data, process implementation and parameterization.

Similarly, DGVM estimates strongly depend on the land use forcing data. For some of the investigated countries, further uncertainties in the DGVM estimates of a similar magnitude are caused by the environmental forcing data used, namely present-day environmental forcing, which is more comparable to BKs and country report-based approaches, compared to



transient environmental forcing, which better reflects historical environmental changes in the real world. In the majority of

investigated countries, $f_{\text{LULUCF}}$ estimates based on official country reports (NGHGI DB and FAOSTAT) are lower compared to the modelled estimates. However, once the varying characteristics and definitions (in particular the so-called managed land proxy) are accounted for, the differences become substantially lower in most countries.

Analysing the gross fluxes from BKs revealed that short-term variations in net fluxes are mostly linked to gross emissions, which show large temporal variability, while gross removals rather impact the long-term trends of net fluxes. Uncertainties in

net fluxes mainly relate to uncertainties in gross emissions (for example in Brazil, Canada, China, and DR Congo) but can also be strongly impacted by uncertainties in gross removals (for example in the USA). In India and Russia, pronounced uncertainties in both gross emissions and gross removals largely compensate each other, which results in rather low uncertainties of the net flux. Furthermore, the investigation of the net-to-gross ratio revealed that the net flux is comprised by large gross fluxes in most countries and over most of the investigated time from 1950 onward. Noteworthy, gross fluxes increasingly compensate

each other in most of the countries over time. Considering only net fluxes might thus miss potentially important and large gross fluxes. In addition, grid cell-wise analysis revealed that pronounced gross and net fluxes may occur within a country at different locations, even though net fluxes are close to zero when averaged at the country-level, which highlights the need for spatially explicit data on gross fluxes.

Consequently, model-based spatial data as presented in this study may support the identification of component-wise, histori-

cal and/or regional "uncertainty hotspots" that particularly need improved $f_{\text{LULUCF}}$ estimations, for example through improved land use forcing data from the 1980s onward in China and for the most recent decades in Brazil and DR Congo. The differences in $f_{\text{LULUCF}}$ estimates in these "hotspots" highlight the need for a careful interpretation of the outputs from the varying methods and for a further reconciliation of the different approaches, in particular regarding the different components considered and the methods used for their estimation. To this end, we argue for a systematic model evaluation and improved parametrization of

models, in particular regarding land use forcing data, parametrized carbon densities, and the different processes represented in the models. Earth observation data, such as from optical satellite measurements (e.g., for carbon densities) and atmospheric inversions (e.g., for carbon fluxes), and their improved incorporation into modeling and country report-based approaches may provide substantial advancements in the assessment and understanding of $CO_2$ fluxes from LULUCF.

## 6 Data availability

The NGHGI DB and the adjusted NGHGI DB data can be found under https://doi.org/10.5281/zenodo.7650360, OSCAR22 output can be found under https://doi.org/10.5281/zenodo.7313498. TRENDY simulations are available via request to S.A.Sitch[at]exeter.ac.uk. A dataset covering all aggregated country data for the period 1950 – 2021 for all datasets used in this study can be found under https://doi.org/10.5281/zenodo.8144174 (Obermeier et al., 2023).



## Appendix A: Description of individual approaches for estimating $f_{LULUCF}$

**A1   Bookkeeping models**

BKs use spatial information on land use activities to derive net $f_{LULUCF}$ by summing up all gross carbon fluxes that occur due to land conversion and land management (Pongratz et al., 2014). To estimate carbon fluxes from LULUCF, BKs rely on observation-based carbon stock densities, growth curves (uptake) and decomposition curves (release) of soil and vegetation carbon that are specific for each conversion type. Fluxes due to land-use conversion can occur instantaneously (fluxes upon or
within the year of LULUCF) and in the years following the conversion (legacy fluxes, e.g. from readjustment of carbon stocks).

Fluxes from peat fire and peat drainage are not directly modelled by BKs but are added from external data. For the GCB2022 simulations, which we use here, peat fire emissions were added from the Global Fire Emission Database (GFED4s; van der Werf et al., 2017) for Brunei Darussalam, Indonesia, Malaysia, and Papua New Guinea. Peat drainage emissions were added for all countries as the average from FAO data (Conchedda and Tubiello, 2020) and DGVM simulations with ORCHIDEE-PEAT
(Qiu et al., 2021) and LPX-Bern (Lienert and Joos, 2018; Müller and Joos, 2021). More details can be found in Friedlingstein et al. (2022b).

The three BKs notably differ in (1) the implemented processes, (2) the land use forcing data, and (3) the assigned carbon densities, response curves, and pool allocation fractions (compare Friedlingstein et al., 2022a; Bastos et al., 2021a):

(1) BLUE22 and OSCAR22 include sub-grid-scale transitions between all vegetation types (e.g. from shifting cultivation – a
rotation cycle between forest and agriculture), which presumably leads to higher emissions, whereas sub-grid-scale transitions are not implemented in H&N22 – only if a country's forest loss reported to FRA exceeds agricultural expansion based on FAO land use data, H&N22 assumes that this area is cleared for shifting cultivation. BLUE22 includes gross fluxes related to degradation from primary to secondary land in case natural vegetation is used as rangeland. H&N22 considers fire management in the USA and southeast Asia, in contrast to BLUE22 and OSCAR22.

(2) BLUE22 uses the LUH2-GCB2022 dataset (an update to most recent harmonized land-use change data (LUH2 v2h); Chini et al., 2021; Hurtt et al., 2020) based on HYDE3.3, whose contemporary land use is constrained by annual ESA CCI Land Cover and updated agricultural areas from FAO (Klein Goldewijk et al., 2017). These data are globally consistent but have a relatively coarse spatial resolution (0.25° x 0.25°) and may thus exclude regional and local specifics (Bastos et al., 2018; Li et al., 2018; Kondo et al., 2022). H&N22 uses FAO-FRA data from 2020 for forest (from 1990 onward and various sources
before; FAO, 2020) and data from FAOSTAT for other land uses, and applies a 5-year running mean on the activity data before flux calculations (Friedlingstein et al., 2022a). OSCAR22 calculates a best-guess estimate of $f_{LULUCF}$ based on a combination of the LUH2-GCB2022 dataset and FAO-FRA data (Gasser et al., 2020, 2022). BLUE22 output is spatially explicit, while H&N22 and OSCAR22 provide country-level estimates.

We further use BLUE data from GCB2019 (hereafter, BLUE19), forced with LUH2-GCB2019 data based on HYDE3.2
(Klein Goldewijk et al., 2011; Chini et al., 2021). As the BLUE model code was not changed between GCB2019 and GCB2022, this allows isolating the impact of changes in the LULUCF forcing data, with HYDE3.2 using one year of ESA CCI as a reference year for the spatial land cover patterns, whereas HYDE3.3 uses time varying ESA maps which led to spatio-





temporally improved land cover maps (Rosan et al., 2021). From 2018 onward, the underlying LUH2 data linearly interpolates the trend in cropland, pasture, and urban area of the previous five years until the year 2021. This approach does not properly reflect the dynamics in regions with intensive LULUCs in most recent years, therefore, the LUH2-GCB2022 land use forcing data in Brazil is taken from the MapBiomas dataset (collection 6) for the 1985 – 2020, according the approach described in Friedlingstein et al. (2022b).

(3) H&N22 assigns vegetation carbon densities at country-level (based on official country reports) and soil carbon densities globally for 20 types of ecosystems (Houghton and Nassikas, 2017). BLUE22 assigns vegetation and soil carbon densities for 11 PFTs globally (based on literature values; Houghton et al., 1983; Hansis et al., 2015). OSCAR22 uses carbon densities derived from DGVMs and response curves specified for 96 world regions and 5 biomes (Gasser et al., 2020, 2022). Similarly, carbon response curves in H&N22 and BLUE22 are assigned for each of the 20 ecosystems and 11 PFTs considered, respectively. The BLUE model, in general, has highest carbon densities implemented which causes high $f_{\text{LULUCF}}$ estimates, as was shown by Bastos et al. 2021a, b, where parameterizing BLUE with H&N carbon densities led to a 24% reduction of global cumulative $f_{\text{LULUCF}}$ from 1850 – 2015. Using carbon densities and response curves that are static over time, BLUE22 and H&N22 do not explicitly model the effects of environmental changes (e.g., increased $CO_2$ concentration and climatic change), although some are implicitly captured within the observed carbon densities and response curves (Pongratz et al., 2014). In contrast, OSCAR22 includes transient environmental response due to its calibration to transient DGVM simulations (Gasser et al., 2017, 2020). In addition, the allocation of HWPs to different product pools differs between the models: BLUE22 uses three HWP pools (with turnover times of 1, 10, and 100 years) with fixed allocation fractions for each PFT. H&N22 assigns time-variant fractions for five pools (fuel, industrial, 1-year, 10-year and 100-year turnover times) specific for each country (Bastos et al., 2021a). OSCAR22 uses three HWP pools (with average turnover times of 0.75, 6.0 and 65 years) and allocation fractions specific to regions and biomes (Gasser et al., 2017, 2020).

In some countries depicted in the appendix, particularly in arid world regions, gross emissions from BKs are negative in some years. This relates to the definition of gross emissions which include emissions from deforestation, forest degradation, wood harvest and fluxes from transitions between natural land, cropland, and pasture. The latter, however, may cause negative carbon fluxes if carbon densities of the initial land cover are lower than the carbon densities of the converted land cover. Consequently, this may cause gross emissions to be negative, particularly in dryland countries with little forest cover. On the global scale this effect is negligible.

## A2 Dynamic global vegetation models

DGVMs are used in the GCB for the uncertainty assessment of $f_{\text{LULUCF}}$ and the estimation of the natural land sink. Moreover, DGVMs are frequently used in detail studies, e.g., on the effects of land-use changes on local climate, due to their implementation of complex biogeochemical and biogeophysical processes and their capacity to simulate transient environmental responses (e.g. (Krause et al., 2018; Winckler et al., 2017; Bright et al., 2017). We aggregated country-level $f_{\text{LULUCF}}$ based on DGVMs that performed simulations within the project "Trends and drivers of the regional-scale emissions and removals of carbon dioxide" (TRENDY; Le Quéré et al. (2014); Sitch et al. (2015), Sitch et al., 2022 *in prep*).





DGVMs do not directly output $f_{\text{LULUCF}}$. Instead, $f_{\text{LULUCF}}$ is estimated as the difference between two simulations with the same environmental forcing, one including and one excluding LULUCF. To derive $f_{\text{LULUCF}}$ for each grid cell and each (yearly) time step, the net biome productivity (NBP) of the simulation including LULUCF is subtracted from the NBP in the simulation excluding LULUCF, the latter using a constant pre-industrial LULUCF map (from 1700) over time. Thereby, $f_{\text{LULUCF}}$ from DGVMs includes instantaneous as well as legacy fluxes, like BK estimates.

DGVM simulations can be forced with different environmental conditions, where some environmental variables are set constant (fixed) at either pre-industrial or present-day levels, or follow observed, transient conditions (for an overview of the different simulations refer to Obermeier et al., 2021). Here, we use simulations under fixed present-day environmental conditions as they most closely resemble BK simulations and country report-based approaches (using observed C densities), and are recommended by RECCAP2 (Ciais et al., 2022). Present-day environmental simulations are run under the $CO_2$ concentration from 2018 throughout the simulated period, and recycle the climate from 1999 – 2018 by using the mean and variability of the individual years in this period (Obermeier et al., 2021). DGVM simulations with present-day conditions are not performed every year. The most recent present-day simulations available stem from TRENDYv8 used in GCB2019 (Friedlingstein et al., 2019).

We additionally employ transient DGVM simulations from TRENDY v8 as transient simulations are operationally available, commonly used within the scientific community and enable us to derive the difference in $f_{\text{LULUCF}}$ that result from different environmental forcing ((by comparing to present-day TRENDYv8 simulations). The transient environmental simulations used here are forced with observation-based temperature, precipitation, and incoming surface radiation data (0.5° x 0.5° resolution) of the Climatic Research Unit (CRU) and the Japanese Reanalysis (JRA; Friedlingstein et al., 2019; Harris et al., 2014), and global atmospheric $CO_2$ concentrations from ice core data before 1958 (Joos and Spahni, 2008) combined with National Oceanic and Atmospheric Administration (NOAA) data from 1958 onward (Dlugokencky and Tans, 2020).

To enable a robust comparison between the different forcings, we selected only those nine DGVMs that provide present-day in addition to transient simulations within TRENDYv8: CLASS-CTEM (Melton and Arora, 2016), DLEM (Tian et al., 2015), JSBACH (Mauritsen et al., 2019), LPJ-GUESS (Smith et al., 2014), LPX-Bern (Lienert and Joos, 2018), ORCHIDEE (Krinner et al., 2005), ORCHIDEE-CNP (Goll et al., 2017), SDGVM (Walker et al., 2017), and VISIT (Kato et al., 2013).

Differences among the DGVMs mainly result from (1) differing model parameters, and (2) the implementation of different processes with varying complexities (Friedlingstein et al., 2022a): (1) Model parametrization differs, for instance, in the distinction of primary and secondary forests and turnover rates of product pools (Kondo et al., 2022), the fraction of directly emitted carbon upon LULUCs, and the implemented decomposition rates and resulting soil carbon densities (Goll et al., 2015). (2) Implemented processes differ, for example, as some DGVMs include fires (without distinguishing whether they are natural or anthropogenic) while others have no fire implemented. Other natural disturbances are not included by the DGVMs. Additionally, some DGVMs consider nitrogen or phosphorus cycles, cropland irrigation, shifting cultivation, forest degradation, residue carbon after deforestation and wood and crop harvests, while others do not (for more details refer to Bastos et al., 2020a; Kondo et al., 2022; Friedlingstein et al., 2022a). Noteworthy, the inclusion of processes, such as shifting cultivation,





wood harvest, grazing, crop harvest, and cropland management increases historic (1901 – 2014) LULUCF emissions from DGVMs by 20 – 30% for each of these processes (Arneth et al., 2017).

## A3 National Greenhouse Gas Inventories under the UNFCCC

To take stock and track progress towards their NDCs, countries report official inventory statistics of GHG emissions and
removals to the UNFCCC via different schemes. In line with Grassi et al. (2023), in this study, we refer to any of such official country reports on anthropogenic GHG data submitted to UNFCCC as National GreenHouse Gas Inventory (NGHGI).

Following the reporting guidelines of UNFCCC, NGHGIs are submitted regularly (annually for Annex I countries and typically at least every few years for Non-Annex I countries). NGHGIs are required to meet the key principles of transparency, accuracy, completeness, consistency, and comparability (TACCC). Nonetheless, depending on national circumstances and con-
ditions, reporting is done with different frequency and sophistication of the underlying methods, and large methodological uncertainties exist (Federici et al., 2017). According to the IPCC best practice guidelines (IPCC, 2006), some flexibility is allowed for forest definitions within the NGHGIs, e.g., thresholds of parameters defining forests (minimum area, tree crown cover, tree height) can be chosen from an allowed interval (0.05 – 1.0 hectares, 10 – 30 percentages, 2 – 5 meters, respectively) in the first national communication, but must be kept constant afterwards. The IPCC defined different tiers that indicate the
sophistication of the applied methods ranging from Tier 1 to Tier 3, with Tier 3 being the most demanding in terms of data availability and method complexity. Most Annex I countries (members of the OECD and some transitional economies) report all land use fluxes annually since 1990 following the 2006 IPCC Guidelines (and partly following the 2019 IPCC refinements) (IPCC, 2006, 2019). Guidelines for Non-Annex I countries (emerging economies, e.g., Brazil, China, Democratic Republic of the Congo, India, Indonesia, Nigeria) are more flexible, the applied methodologies are generally less complex, and reporting
often started in 2000 only (IPCC, 2006).

The most important developing countries (Brazil, Indonesia, China, India, Mexico) rely on national inventories, and in the case of Brazil even have a NGHGI comparable to some of the developed countries. In contrast, NGHGIs from most other Non-Annex I countries rely on empirical emission factors to estimate $f_{\mathrm{LULUCF}}$, which are representative rates of emissions (e.g. for specific forest and climate types) that are usually obtained from averaged measurement data sampled under certain
environmental conditions, and thus, hardly capture local dynamics. Such basic approaches using default values correspond to Tier 1 level, while reports from Annex I countries are often based on country-specific statistical or process-based models using national activity data (i.e., Tier 2 or Tier 3 level).

To reduce the uncertainties associated with the different reporting schemes, we use the newly compiled data from Grassi et al. (2023), which is based on official country reports to the UNFCCC, but additionally includes quality checks, and gap-filling if
necessary (hereafter named NGHGI DB). NGHGI DB data is freely available under https://doi.org/10.5281/zenodo.7650360 from 2000 onward and is considered a realistic approximation of the data to be used for the forthcoming Global Stocktake in 2023 (Grassi et al., 2023).

NGHGIs are supposed to encompass all LULUCF fluxes from areas considered managed, including forest land, cropland, grassland, wetlands, settlements, and other land, as well as emissions from organic soils and fires (and some also include



shifting cultivation). Thereby, the reports should include pools for dead wood, litter, soil organic C, and HWPs. However, many Non-Annex I countries report only fluxes from deforestation and only few include fluxes from other LULUCF categories. To distinguish between anthropogenic and non-anthropogenic fluxes, national authorities use the "managed land" proxy where emissions and removals on managed lands are counted for, whereas fluxes on unmanaged lands are not reported.

## A4  FAOSTAT

The FAOSTAT $f_{\text{LULUCF}}$ data are a component of the FAO Emissions database, developed to assess the role of food and agriculture in global anthropogenic emissions (Tubiello, 2019; Tubiello et al., 2022; FAO, 2021). The FAOSTAT $f_{\text{LULUCF}}$ data cover carbon emissions and carbon removals as well as non-$CO_2$ emissions from biomass fires (not considered in this work) on the following IPCC land use and land use change categories: (1) emissions and removals on forests and from deforestation (Tubiello et al., 2021b), (2) emissions from peatland fires (Rossi et al., 2016; Prosperi et al., 2020), and (3) emissions from 675 peatland drainage (Conchedda and Tubiello, 2020; Tubiello et al., 2016).

FAOSTAT $f_{\text{LULUCF}}$ data cover 238 countries and territories, with sub-regional, regional and global aggregates for the period 1990 – 2020. They are estimated by applying IPCC (2006) guidelines to LULUCF activity data generated either through official country reporting processes, such as the FRA (FAO, 2020), or through analysis of geospatial data carried out by FAO under its mandate. Specifically, forest fluxes are based on carbon stock change statistics computed directly at national level, 680 following carbon stock and (net) area change statistics reported by member countries to FAO at five-year intervals (FAO, 2020; Tubiello et al., 2021b). Deforestation emissions are computed separately for the two FAO forest types, *naturally regenerating forest* (comprising primary and secondary natural forests) and *planted forest*. Noteworthy, FAOSTAT estimates include only the above- and below-ground biomass pools, while fluxes resulting from changes in other pools, such as the soil carbon pools, are not modelled, except for those in organic soils. The estimated emissions and removals on forest land do not distinguish 685 between anthropogenic and non-anthropogenic fluxes, and include indirect climate and $CO_2$ effects in line with the IPCC management land proxy. Additionally, FAOSTAT $f_{\text{LULUCF}}$ data comprises anthropogenic peat fires computed at grid cell level using the histosols map from the Harmonized World Soils Database (as proxy for spatial peatland distribution) combined with remote sensing products of MODIS burnt area and underlying MODIS land cover maps. IPCC (2013) tier 1 emission factors are associated to each land cover/ecological zone by merging with the IPCcarbon-JRC agro-climatic zone map (FAOSTAT, 690 2020) and results are aggregated at country level (for details, refer to Rossi et al., 2016; Prosperi et al., 2020). Similarly as for the BKs, peat fire emissions are only considered anthropogenic for Southeast Asian countries albeit data is available globally. Emissions from drained peatlands, computed also at grid cell level, are estimated based on the harmonized world map of histosols and the ESA CCI land cover map to identify cropland areas, assuming cultivation on peatland area is a proxy for anthropogenic drainage (Conchedda and Tubiello, 2020).





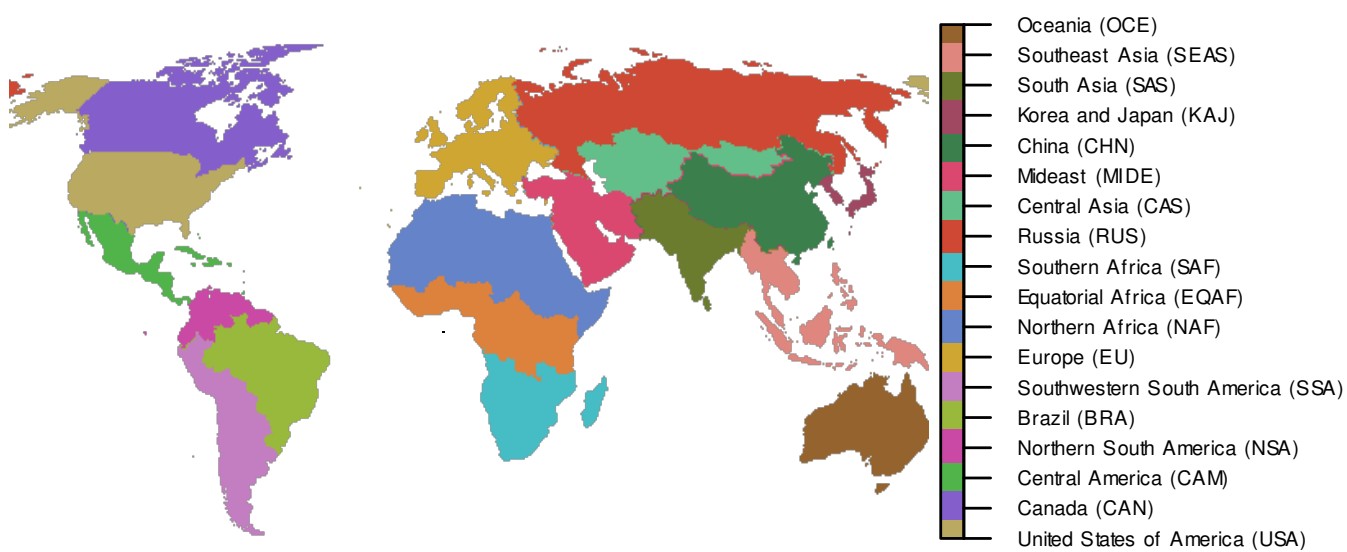

**Figure A1.** Regions as defined by 'REgional Carbon Cycle Assessment and Processes' Phase 2 (RECCAP; Tian et al., 2019)

.

**Figure A2.** Net per-capita carbon fluxes from land use, land-use change and forestry ($f_{\text{LULUCF}}$) from three bookkeeping models (BKs, data from GCB2022 simulations). (a) Cumulative per-capita carbon fluxes over 1950 – 2021 and (b) average per-capita carbon fluxes in 2011 – 2021. The bars show the mean of the three BKs (filled bars), and minimum and maximum estimates (hatched bars). Numbers in parantheses show the multi-model average and standard deviation (in tC per capita in (a) and tC per capita yr$^{-1}$ in (b)). Colors indicate the absolute quantities, showing countries with net emissions in red and countries with net removals in green. All 186 country aggregates from this study are shown in decreasing order of their (a) cumulative and (b) most recent annual $f_{\text{LULUCF}}$. In each panel, the top ten emitters and the five countries with the largest removals are labelled. The figure corresponds to Fig. 1 in the main manuscript with the difference that fluxes are shown per capita. Note that values for very small countries should be interpreted with care as the relatively low resolution of many models creates uncertainty at the small scale.

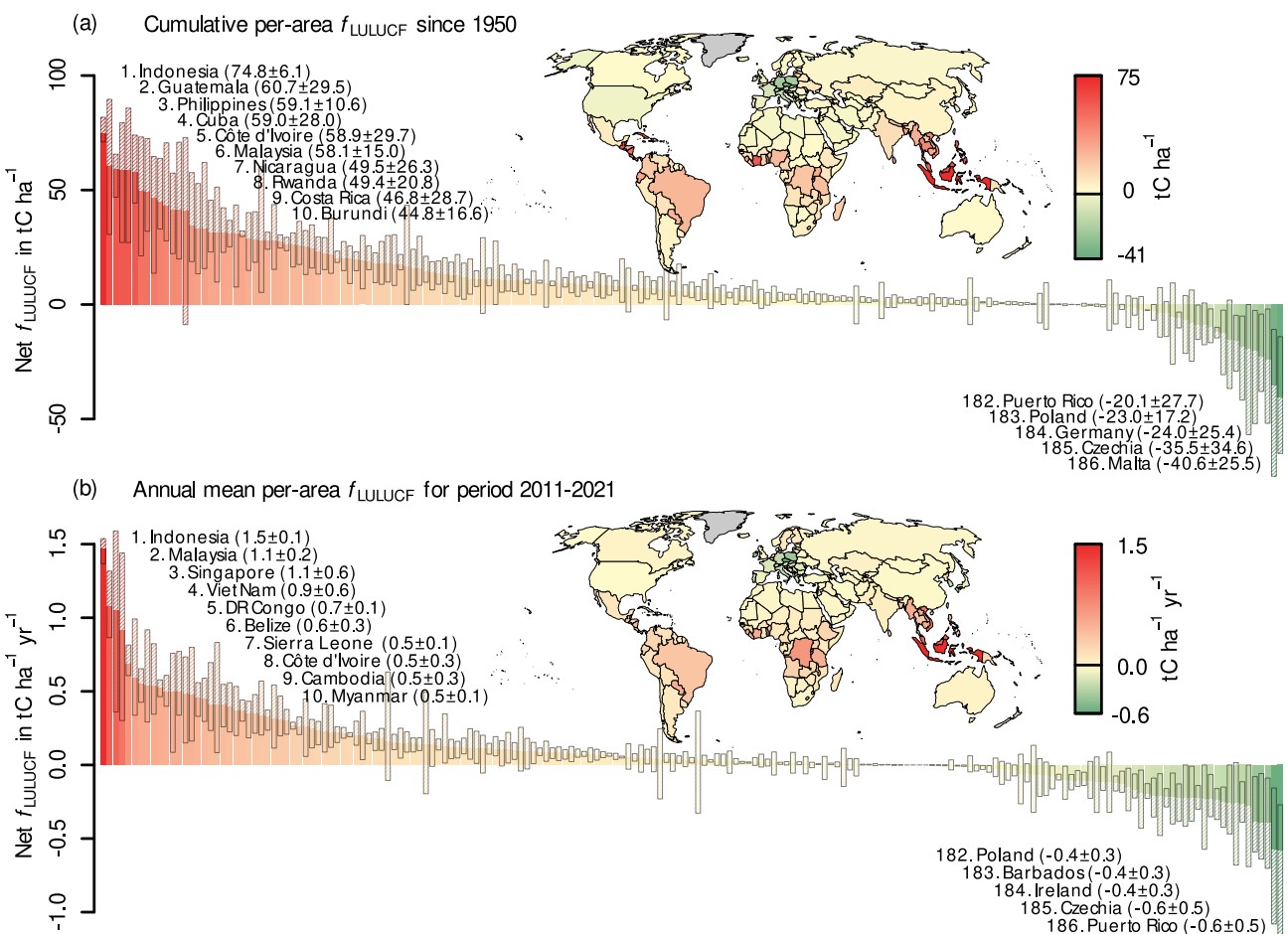

**Figure A3.** Net per-area carbon fluxes from land use, land-use change and forestry ($f_{\text{LULUCF}}$) from three bookkeeping models (BKs, data from GCB2022 simulations). (a) Cumulative per-area carbon fluxes over 1950 – 2021 and (b) average per-area carbon fluxes in 2011 – 2021. The bars show the mean of the three BKs (filled bars), and minimum and maximum estimates (hatched bars). Numbers in parantheses show the multi-model average and standard deviation (in tC ha$^{-1}$ in (a) and tC ha$^{-1}$ yr$^{-1}$ in (b)). Colors indicate the absolute quantities, showing countries with net emissions in red and countries with net removals in green. All 186 country aggregates from this study are shown in decreasing order of their (a) cumulative and (b) most recent annual $f_{\text{LULUCF}}$. In each panel, the top ten emitters and the five countries with the largest removals are labelled. The figure corresponds to Fig. 1 in the main manuscript with the difference that the fluxes are shown per area. Note that values for very small countries should be interpreted with care as the relatively low resolution of many models creates uncertainty at the small scale.

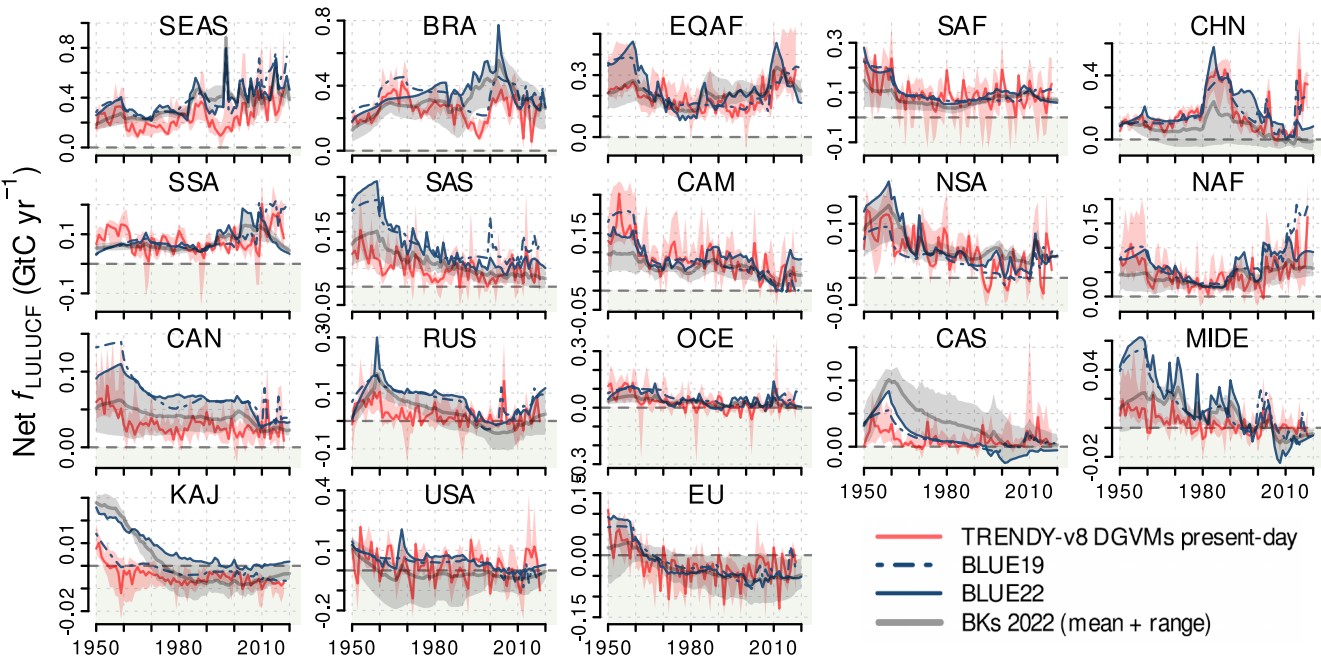

**Figure A4.** Time series of net carbon flux from land use, land-use change and forestry ($f_{\text{LULUCF}}$) in 1950 – 2021 derived by bookkeeping models (BKs) and TRENDYv8 simulations with dynamic global vegetation models (DGVMs; used in the GCB2019) under present-day climate forcing for the RECCAP2 regions. Regions are sorted according to their cumulative net emissions from 1950 – 2021 as derived by three bookkeeping models. The complete region designations for the used acronyms can be found in the legend of Figure A1. The figure shows the mean and absolute range of three BKs (using the GCB2022 simulations, BKs 2022) and the median and interquartile range of the eight DGVMs. Additionally, estimates from BLUE simulations from the GCB2019 (BLUE19; blue solid) and from the GCB2022 (BLUE22; blue dashed) are shown to illustrate the impact of updates in the LUHv2 forcing data. BLUE19 data is only available until 2019 and TRENDYv8 data are only available until 2018. Greenish background depicts negative $f_{\text{LULUCF}}$, that is carbon removal from the atmosphere. The figure corresponds to Fig. 2 in the main manuscript with the difference that the fluxes are shown for the RECCAP regions.

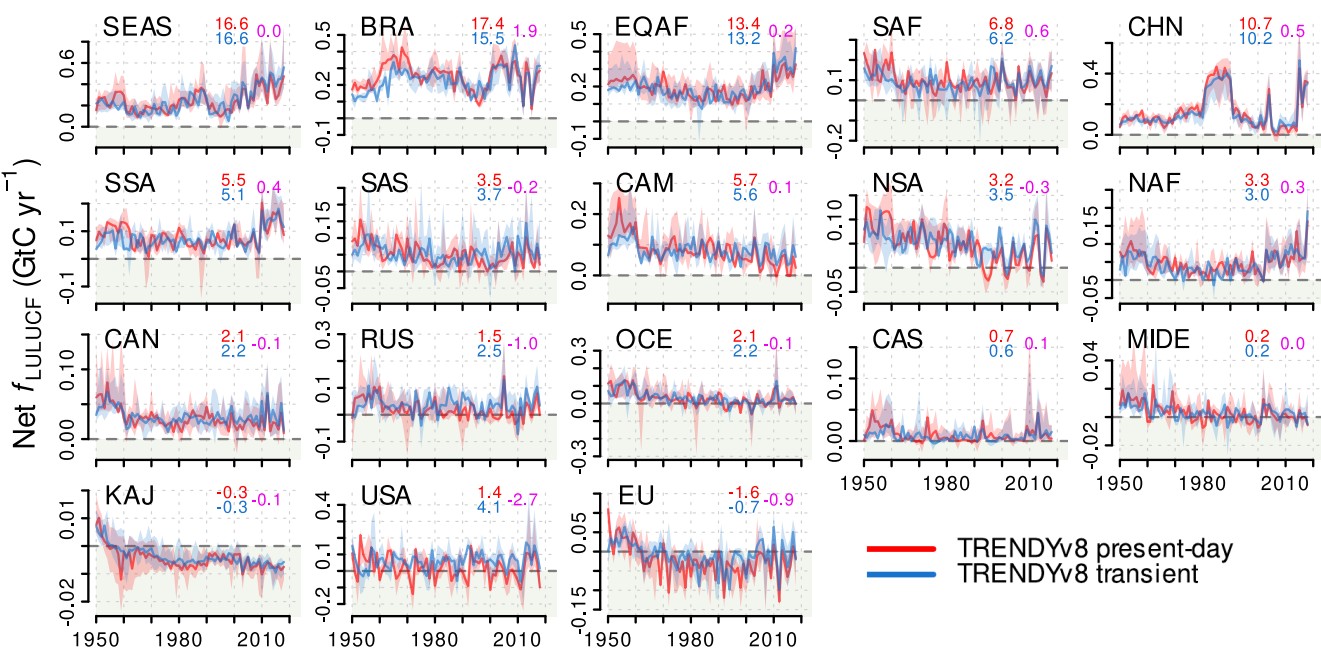

**Figure A5.** Time series of net carbon flux from land use, land-use change and forestry ($f_{LULUCF}$) in $1950 - 2021$ derived from TRENDYv8 simulations with dynamic global vegetation models (DGVMs) under historical (transient) and fixed present-day environmental conditions (compare appendix) for the RECCAP2 regions. Regions are sorted according to their cumulative net emissions from $1950 - 2021$ as derived by three bookkeeping models. The complete region designations for the used acronyms can be found in the legend of Figure A1. The figure shows the median and interquartile range of the eight DGVMs. Greenish background depicts negative $f_{LULUCF}$, that is carbon removal from the atmosphere. The figure corresponds to Fig. 3 in the main manuscript with the difference that the fluxes are shown for the RECCAP regions.

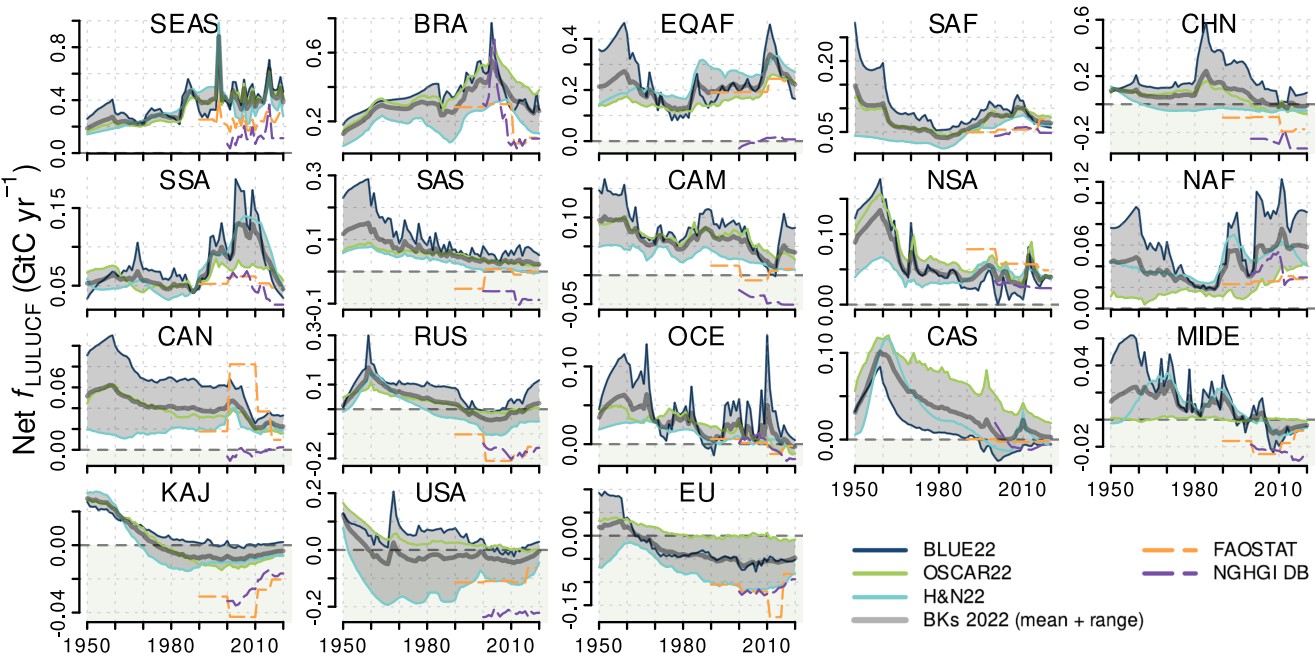

**Figure A6.** Net carbon flux from land use, land-use change and forestry ($f_{\mathrm{LULUCF}}$) in 1950 – 2019 derived from individual bookkeeping models (BKs) used in GCB2022 (BKs 2022) and country report-based estimates (FAOSTAT and NGHGI DB) for the RECCAP2 regions. The use of dashed and solid lines indicates that BK estimates and country report-based estimates are not directly comparable (see Sect. 2.5 and the appendix). For better comparability, the adjusted NGHGI DB estimates, matching the $f_{\mathrm{LULUCF}}$ definition of the BKs, are additionally shown. The gray line (shading) depicts the median (range) of the three BKs. The light green background indicates negative $f_{\mathrm{LULUCF}}$ that is net carbon removal from the atmosphere by LULUCF. The figure corresponds to Fig. 4 in the main manuscript with the difference that the fluxes are shown for the RECCAP regions.

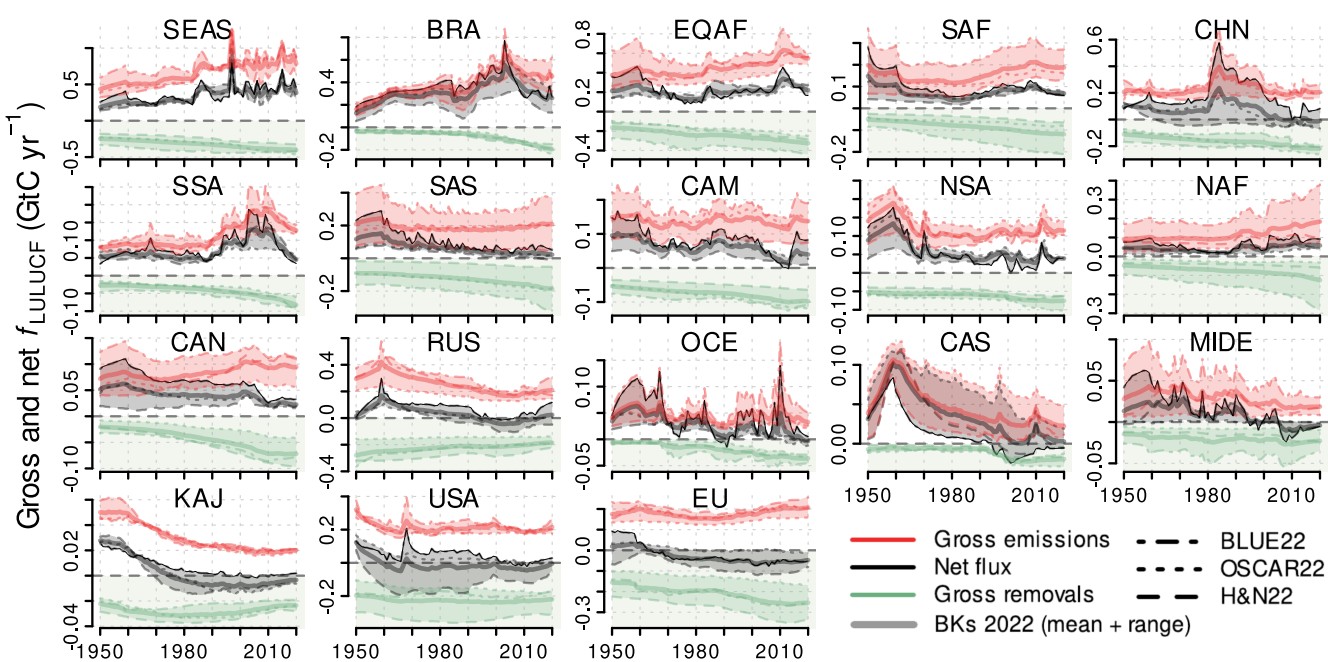

**Figure A7.** Gross and net fluxes from land use, land-use change and forestry ($f_{\mathrm{LULUCF}}$) from 1950 until 2021 derived by three bookkeeping models (BKs; used in the GCB2022) for the RECCAP2 regions. Regions are sorted according to their cumulative net emissions from 1950 – 2021 as derived by three bookkeeping models. The complete region designations for the used acronyms can be found in the legend of Figure A1. Greenish background depicts negative $f_{\mathrm{LULUCF}}$, that is carbon removal from the atmosphere. The figure corresponds to Fig. 5 in the main manuscript with the difference that the fluxes are shown for the RECCAP regions.

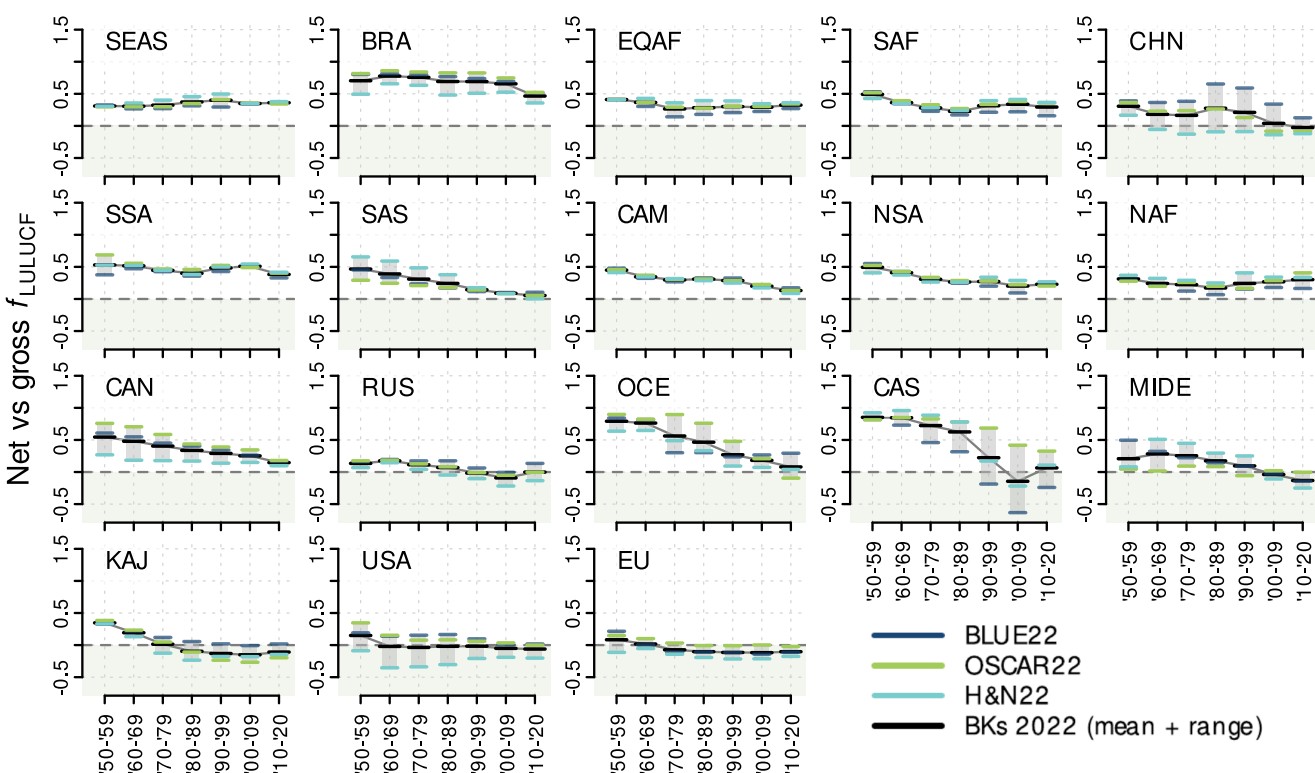

**Figure A8.** Gross and net fluxes from land use, land-use change and forestry ($f_{\text{LULUCF}}$) from 1950 until 2021 derived by three bookkeeping models (BKs; used in the GCB2022) for the RECCAP2 regions. Regions are sorted according to their cumulative net emissions from 1950 – 2021 as derived by three bookkeeping models. The complete region designations for the used acronyms can be found in the legend of Figure A1. Lines (shaded area) depict the mean (range) of the BK estimates. The figure corresponds to Fig. 6 in the main manuscript with the difference that the fluxes are shown for the RECCAP regions.



**Figure A9.** As in Figure 2 but for the 93 top emitting countries according to their cumulative emission in 1950-2021 as derived by three bookkeeping models (without the lines for the differences of the estimations; remaining countries are shown in Figure A10). Green shaded areas depict the range of net carbon removals. For the complete country designations of the Alpha-3 codes, refer to Tables A1-A3.



**Figure A10.** As in Figure A9 but showing the lowest emitting countries and countries with net removals according to their cumulative fluxes in 1950-2021 as derived by three bookkeeping models. Green shaded areas depict the range of net carbon removals. For the official country names and each country's rank (order of subplots), refer to Tables A1-A3.

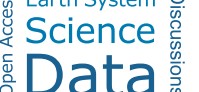

**Figure A11.** As in Figure 3 but for the 93 top emitting countries according to their cumulative emission in 1950-2021 as derived by three bookkeeping models (without the lines for the differences of the estimations; remaining countries are shown in Figure A12). Green shaded areas depict the range of net carbon removals. For the official country names and each country's rank (order of subplots), refer to Tables A1-A3.





**Figure A12.** As in Figure A11 but showing the lowest emitting countries and countries with net removals according to their cumulative fluxes in 1950-2020 as derived by three bookkeeping models. Green shaded areas depict the range of net carbon removals. For the official country names and each country's rank (order of subplots), refer to Tables A1-A3.



**Figure A13.** As in Figure 4 but for the 93 top emitting countries according to their cumulative emission in 1950-2020 as derived by three bookkeeping models (remaining countries are shown in Figure A14). Green shaded areas depict the range of net carbon removals. For the official country names and each country's rank (order of subplots), refer to Tables A1-A3.



**Figure A14.** As in Figure A13 but showing the lowest emitting countries and countries with net removals according to their cumulative fluxes in 1950-2020 as derived by three bookkeeping models. Green shaded areas highlight net carbon removals. For the official country names and each country's rank (order of subplots), refer to Tables A1-A3.





**Figure A15.** As in Figure 5 but for the 93 top emitting countries according to their cumulative emission in 1950-2021 as derived by three bookkeeping models (remaining countries are shown in Figure A15). Green shaded areas highlight net carbon removals. For the official country names and each country's rank (order of subplots), refer to Tables A1-A3.



**Figure A16.** As in Figure A15 but showing the lowest emitting countries and countries with net removals according to their cumulative fluxes in 1950-2021 as derived by three bookkeeping models. Green shaded areas highlight net carbon removals. For the official country names and each country's rank (order of subplots), refer to Tables A1-A3.



**Figure A17.** As in Figure 6 but for the 93 top emitting countries according to their cumulative emission in 1950-2021 as derived by three bookkeeping models (remaining countries are shown in Figure A18. Green shaded areas highlight net carbon removals. For the official country names and each country's rank (order of subplots), refer to Tables A1-A3.



**Figure A18.** As in Figure A17 but showing the lowest emitting countries and countries with net removals according to their cumulative fluxes in 1950-2020 as derived by three bookkeeping models. Green shaded areas highlight of net carbon removals. For the official country names and each country's rank (order of subplots), refer to Tables A1-A3.





**Table A1.** Mean, range, minimum and maximum estimates of country-level annual mean (2011 – 2021) and cumulative (1950 – 2021) net $f_{\text{LULUCF}}$ as derived from three bookkeeping models, plus the respective country's rank. Countries are sorted in alphabetical order. Note, the rank of the cumulative $f_{\text{LULUCF}}$ (in bold) gives the order of countries as they occur in Figs. A9 – A18.

| Country | Code | Cumulative $f_{\text{LULUCF}}$ in 1950 – 2021 (GtC | | | | | Annual mean $f_{\text{LULUCF}}$ in 2011 – 2021 (MtC yr$^{-1}$) | | | | |
| | | Mean | Range | Min | Max | **Rank** | Mean | Range | Min | Max | Rank |
|---|---|---|---|---|---|---|---|---|---|---|---|
| Afghanistan | AFG | 0.095 | 0.096 | 0.048 | 0.145 | **83** | 0.336 | 0.719 | 0.094 | 0.813 | 95 |
| Albania | ALB | 0.017 | 0.075 | −0.018 | 0.057 | **119** | 0.016 | 0.314 | −0.164 | 0.15 | 118 |
| Algeria | DZA | 0.074 | 0.287 | −0.082 | 0.205 | **88** | 0.063 | 1.49 | −0.821 | 0.669 | 112 |
| Andorra | AND | 0 | 0.075 | 0 | 0.001 | **147** | 0 | 0.314 | 0 | 0 | 128 |
| Angola | AGO | 1.302 | 0.881 | 0.733 | 1.613 | **20** | 32.122 | 16.523 | 24.005 | 40.527 | 7 |
| Argentina | ARG | 1.446 | 1.686 | 0.47 | 2.156 | **18** | 22.116 | 35.909 | 2.262 | 38.171 | 12 |
| Armenia | ARM | 0.029 | 0.03 | 0.012 | 0.042 | **109** | 0.311 | 0.61 | 0.025 | 0.635 | 97 |
| Australia | AUS | 1.742 | 1.881 | 1.044 | 2.925 | **17** | 11.341 | 20.364 | 2.734 | 23.097 | 22 |
| Austria | AUT | −0.151 | 0.303 | −0.344 | −0.042 | **180** | −1.725 | 3.421 | −3.921 | −0.5 | 167 |
| Azerbaijan | AZE | 0.075 | 0.076 | 0.042 | 0.118 | **87** | 0.912 | 1.137 | 0.527 | 1.665 | 71 |
| Bahamas, The | BHS | 0.001 | 0.001 | 0.001 | 0.001 | **140** | −0.007 | 1.095 | −0.01 | 0 | 136 |
| Bahrain | BHR | 0 | 0.14 | 0 | 0 | **152** | 0 | 1.095 | 0 | 0 | 129 |
| Bangladesh | BGD | 0.408 | 0.253 | 0.296 | 0.55 | **43** | 4.122 | 2.193 | 2.926 | 5.119 | 44 |
| Barbados | BRB | 0 | 0.001 | 0 | 0.001 | **145** | −0.018 | 0.025 | −0.028 | −0.004 | 140 |
| Belarus | BLR | 0.15 | 0.044 | 0.126 | 0.17 | **71** | −0.485 | 2.445 | −1.489 | 0.955 | 158 |
| Belgium | BEL | −0.011 | 0.065 | −0.043 | 0.022 | **165** | 0.58 | 0.339 | 0.369 | 0.708 | 80 |
| Belize | BLZ | 0.059 | 0.043 | 0.04 | 0.083 | **92** | 1.383 | 1.518 | 0.79 | 2.308 | 66 |
| Benin | BEN | 0.288 | 0.217 | 0.148 | 0.365 | **51** | 4.536 | 3.156 | 2.924 | 6.08 | 39 |
| Bhutan | BTN | 0.107 | 0.209 | 0.02 | 0.23 | **80** | 0.568 | 2.727 | −0.4 | 2.327 | 81 |
| Bolivia | BOL | 1.127 | 0.2 | 0.997 | 1.198 | **22** | 15.175 | 11.079 | 8.732 | 19.811 | 18 |
| Bosnia and Herzegovina | BIH | 0.03 | 0.026 | 0.018 | 0.044 | **108** | 0.081 | 0.758 | −0.303 | 0.455 | 109 |
| Botswana | BWA | 0.103 | 0.141 | 0.054 | 0.195 | **81** | 2.326 | 5.765 | 0.286 | 6.052 | 55 |
| Brazil | BRA | 21.801 | 12.461 | 13.702 | 26.163 | **1** | 285.283 | 222.921 | 170.548 | 393.469 | 1 |
| British Virgin Islands | VGB | 0 | 0.606 | 0 | 0.001 | **149** | 0 | 11.413 | 0 | 0 | 131 |
| Brunei Darussalam | BRN | 0.014 | 0.003 | 0.013 | 0.016 | **123** | 0.233 | 0.176 | 0.14 | 0.316 | 104 |
| Bulgaria | BGR | 0.002 | 0.14 | −0.087 | 0.053 | **137** | −1.131 | 1.095 | −1.542 | −0.446 | 164 |
| Burkina Faso | BFA | 0.243 | 0.078 | 0.197 | 0.274 | **57** | 3.342 | 1.729 | 2.582 | 4.311 | 49 |
| Burundi | BDI | 0.115 | 0.079 | 0.085 | 0.164 | **78** | 0.995 | 0.878 | 0.55 | 1.428 | 70 |
| Cambodia | KHM | 0.557 | 0.731 | 0.174 | 0.905 | **38** | 9.836 | 8.741 | 6.495 | 15.236 | 26 |
| Cameroon | CMR | 0.695 | 0.535 | 0.461 | 0.996 | **31** | 8.06 | 6.708 | 4.554 | 11.262 | 30 |
| Canada | CAN | 2.939 | 3.345 | 1.4 | 4.746 | **6** | 23.225 | 15.395 | 17.079 | 32.475 | 11 |
| Central African Republic | CAF | 0.191 | 0.139 | 0.139 | 0.278 | **64** | 1.968 | 2.295 | 0.84 | 3.135 | 59 |
| Chad | TCD | 0.212 | 0.08 | 0.176 | 0.255 | **60** | 8.53 | 5.153 | 6.365 | 11.517 | 29 |
| Chile | CHL | 0.367 | 0.467 | 0.082 | 0.549 | **46** | 3.243 | 13.698 | −3.688 | 10.01 | 53 |
| China | CHN | 4.787 | 14.325 | −1.675 | 12.649 | **3** | −9.03 | 110.042 | −46.485 | 63.557 | 184 |
| Colombia | COL | 2.211 | 0.919 | 1.64 | 2.559 | **9** | 25.223 | 10.831 | 20.048 | 30.879 | 10 |
| Congo, Dem. Rep. | COD | 4.614 | 1.125 | 4.169 | 5.293 | **4** | 155.269 | 40.147 | 140.109 | 180.256 | 3 |
| Congo, Rep. | COG | 0.356 | 0.373 | 0.152 | 0.526 | **47** | 3.937 | 3.546 | 2.566 | 6.113 | 45 |
| Cook Islands | COK | 0 | 0.373 | 0 | 0 | **153** | −0.005 | 3.546 | −0.01 | 0 | 134 |
| Costa Rica | CRI | 0.239 | 0.26 | 0.07 | 0.33 | **58** | 0.335 | 0.374 | 0.147 | 0.521 | 96 |
| Côte d'Ivoire | CIV | 1.872 | 1.868 | 0.86 | 2.728 | **13** | 17.281 | 19.955 | 9.61 | 29.565 | 15 |
| Croatia | HRV | −0.003 | 0.075 | −0.047 | 0.028 | **161** | −0.598 | 0.62 | −0.999 | −0.379 | 162 |
| Cuba | CUB | 0.612 | 0.543 | 0.281 | 0.825 | **35** | −1.715 | 1.611 | −2.314 | −0.703 | 166 |
| Cyprus | CYP | −0.011 | 0.004 | −0.013 | −0.009 | **166** | −0.187 | 0.105 | −0.249 | −0.144 | 151 |
| Czech Republic | CZE | −0.274 | 0.495 | −0.579 | −0.084 | **181** | −4.471 | 6.98 | −8.198 | −1.218 | 178 |
| Denmark | DNK | 0.008 | 0.066 | −0.034 | 0.032 | **126** | 0.253 | 0.316 | 0.062 | 0.378 | 103 |
| Djibouti | DJI | 0 | 0.003 | −0.002 | 0.001 | **156** | 0.005 | 0.029 | −0.009 | 0.02 | 122 |
| Dominica | DMA | 0.001 | 0.002 | 0 | 0.002 | **141** | −0.005 | 0.033 | −0.024 | 0.009 | 135 |
| Dominican Republic | DOM | 0.138 | 0.107 | 0.089 | 0.196 | **74** | −0.516 | 0.447 | −0.796 | −0.349 | 159 |
| Ecuador | ECU | 0.786 | 0.562 | 0.57 | 1.132 | **29** | 5.747 | 3.781 | 3.509 | 7.29 | 35 |
| Egypt, Arab Rep. | EGY | 0.117 | 0.266 | 0.012 | 0.278 | **77** | 0.82 | 1.946 | 0.144 | 2.09 | 74 |
| El Salvador | SLV | 0.069 | 0.096 | 0.033 | 0.129 | **89** | −0.258 | 0.905 | −0.727 | 0.177 | 154 |
| Equatorial Guinea | GNQ | 0.049 | 0.019 | 0.043 | 0.061 | **95** | 0.547 | 0.892 | 0.115 | 1.006 | 82 |
| Eritrea | ERI | 0.017 | 0.023 | 0.007 | 0.03 | **120** | 0.484 | 0.805 | 0.201 | 1.006 | 86 |
| Estonia | EST | −0.03 | 0.108 | −0.091 | 0.017 | **173** | −0.089 | 0.661 | −0.518 | 0.143 | 144 |
| Ethiopia | ETH | 0.995 | 1.202 | 0.392 | 1.595 | **26** | 25.452 | 38.049 | 9.761 | 47.81 | 9 |
| Fiji | FJI | 0.029 | 0.039 | 0.015 | 0.054 | **110** | 0.074 | 0.307 | −0.08 | 0.227 | 111 |
| Finland | FIN | 0.108 | 0.253 | −0.051 | 0.202 | **79** | 3.298 | 2.368 | 2.195 | 4.563 | 51 |
| France | FRA | −0.491 | 0.825 | −0.836 | −0.011 | **183** | −7.948 | 15.478 | −16.857 | −1.379 | 182 |
| French Guiana | GUF | 0.018 | 0.014 | 0.013 | 0.026 | **118** | 0.466 | 0.318 | 0.327 | 0.645 | 89 |
| French Polynesia | PYF | 0 | 0 | 0 | 0 | **155** | −0.013 | 8.934 | −0.025 | 0 | 138 |
| Gabon | GAB | 0.18 | 0.151 | 0.118 | 0.269 | **68** | 1.916 | 1.441 | 1.373 | 2.814 | 60 |



**Table A2.** Mean, range, minimum and maximum estimates of country-level annual mean (2011 – 2021) and cumulative (1950 – 2021) net $f_{LULUCF}$ as derived from three bookkeeping models, plus the respective country's rank. Countries are sorted in alphabetical order. Note, the rank of the cumulative $f_{LULUCF}$ (in bold) gives the order of countries as they occur in Figs. A9 – A18.

| Country | Code | Cumulative $f_{LULUCF}$ in 1950 – 2021 (GtC) | | | | | Annual mean $f_{LULUCF}$ in 2011 – 2021 (MtC yr$^{-1}$) | | | | |
|---|---|---|---|---|---|---|---|---|---|---|---|
| | | Mean | Range | Min | Max | **Rank** | Mean | Range | Min | Max | Rank |
| Gambia, The | GMB | 0.018 | 0.021 | 0.01 | 0.03 | **117** | 0.381 | 0.709 | 0.056 | 0.765 | 94 |
| Georgia | GEO | −0.106 | 0.268 | −0.255 | 0.014 | **178** | −1.615 | 1.435 | −2.54 | −1.105 | 165 |
| Germany | DEU | −0.84 | 1.744 | −1.812 | −0.068 | **185** | −8.779 | 24.136 | −23.653 | 0.484 | 183 |
| Ghana | GHA | 0.402 | 0.504 | 0.19 | 0.694 | **45** | 2.228 | 5.142 | 0.256 | 5.398 | 57 |
| Greece | GRC | 0.019 | 0.033 | 0.008 | 0.042 | **113** | −2.504 | 5.321 | −5.786 | −0.465 | 173 |
| Guadeloupe | GLP | 0.003 | 0.003 | 0.001 | 0.004 | **133** | 0.024 | 0.12 | −0.032 | 0.088 | 115 |
| Guatemala | GTM | 0.651 | 0.632 | 0.328 | 0.96 | **32** | 3.347 | 1.761 | 2.313 | 4.074 | 48 |
| Guinea | GIN | 0.257 | 0.295 | 0.139 | 0.433 | **54** | 4.416 | 1.741 | 3.521 | 5.262 | 40 |
| Guinea-Bissau | GNB | 0.038 | 0.029 | 0.025 | 0.053 | **102** | 0.395 | 0.581 | 0.11 | 0.691 | 92 |
| Guyana | GUY | 0.18 | 0.039 | 0.162 | 0.201 | **69** | 1.657 | 1.019 | 1.288 | 2.307 | 64 |
| Haiti | HTI | 0.06 | 0.066 | 0.038 | 0.104 | **91** | 0.799 | 0.765 | 0.518 | 1.284 | 75 |
| Honduras | HND | 0.346 | 0.401 | 0.086 | 0.486 | **48** | 2.525 | 3.306 | 1.374 | 4.68 | 54 |
| Hungary | HUN | −0.083 | 0.166 | −0.16 | 0.006 | **176** | −2.095 | 2.605 | −3.213 | −0.607 | 169 |
| Iceland | ISL | 0.011 | 0.018 | 0 | 0.018 | **125** | 0.038 | 0.074 | 0.01 | 0.084 | 114 |
| India | IND | 3.299 | 4.553 | 1.35 | 5.903 | **5** | 15.349 | 52.845 | −9.659 | 43.186 | 17 |
| Indonesia | IDN | 14.038 | 2.022 | 13.338 | 15.36 | **2** | 283.084 | 31.705 | 265.159 | 296.864 | 2 |
| Iran, Islamic Rep. | IRN | 0.123 | 0.236 | −0.012 | 0.224 | **75** | −2.418 | 6.592 | −6.662 | −0.07 | 172 |
| Iraq | IRQ | 0.045 | 0.108 | −0.011 | 0.097 | **96** | 1.361 | 3.066 | −0.133 | 2.934 | 67 |
| Ireland | IRL | −0.084 | 0.194 | −0.211 | −0.017 | **177** | −2.767 | 4.085 | −4.834 | −0.748 | 174 |
| Israel | ISR | 0.007 | 0.012 | −0.001 | 0.011 | **127** | 0.258 | 0.415 | −0.016 | 0.398 | 102 |
| Italy | ITA | −0.47 | 0.72 | −0.859 | −0.139 | **182** | −7.719 | 7.932 | −11.436 | −3.505 | 181 |
| Jamaica | JAM | 0.03 | 0.031 | 0.017 | 0.047 | **107** | 0.002 | 0.717 | −0.36 | 0.357 | 125 |
| Japan | JPN | 0.037 | 0.74 | −0.319 | 0.422 | **103** | −3.972 | 5.718 | −5.977 | −0.259 | 176 |
| Jordan | JOR | 0.005 | 0.013 | −0.001 | 0.012 | **129** | 0.093 | 0.124 | 0.036 | 0.16 | 108 |
| Kazakhstan | KAZ | 1.823 | 2.681 | 0.572 | 3.254 | **14** | 7.366 | 26.369 | −3.122 | 23.247 | 31 |
| Kenya | KEN | 0.509 | 0.48 | 0.202 | 0.683 | **40** | 4.149 | 3.425 | 2.46 | 5.885 | 43 |
| Korea, Dem. People's Rep. | PRK | 0.101 | 0.129 | 0.034 | 0.163 | **82** | 1.707 | 1.679 | 1.02 | 2.699 | 63 |
| Korea, Rep. | KOR | −0.005 | 0.214 | −0.109 | 0.105 | **163** | −2.161 | 1.595 | −2.945 | −1.35 | 170 |
| Kuwait | KWT | 0 | 0.001 | 0 | 0.001 | **150** | 0.002 | 1.595 | 0 | 0.005 | 126 |
| Kyrgyz Republic | KGZ | 0.079 | 0.128 | 0.031 | 0.158 | **86** | 0.464 | 1.514 | −0.053 | 1.461 | 90 |
| Lao PDR | LAO | 0.441 | 0.538 | 0.158 | 0.696 | **42** | 10.97 | 13.061 | 3.449 | 16.51 | 23 |
| Latvia | LVA | 0.018 | 0.026 | 0.002 | 0.028 | **116** | 0.481 | 0.575 | 0.21 | 0.785 | 87 |
| Lebanon | LBN | 0.003 | 0.004 | 0.001 | 0.005 | **134** | 0.013 | 0.075 | −0.012 | 0.063 | 120 |
| Lesotho | LSO | 0.019 | 0.02 | 0.011 | 0.031 | **115** | 0.385 | 1.036 | 0.034 | 1.07 | 93 |
| Liberia | LBR | 0.193 | 0.122 | 0.143 | 0.265 | **63** | 4.374 | 2.211 | 3.161 | 5.372 | 41 |
| Libya | LBY | −0.014 | 0.098 | −0.079 | 0.019 | **168** | −0.226 | 0.164 | −0.284 | −0.12 | 153 |
| Liechtenstein | LIE | 0 | 0.002 | 0 | 0 | **154** | −0.004 | 0.015 | −0.01 | 0 | 133 |
| Lithuania | LTU | 0.081 | 0.054 | 0.062 | 0.116 | **84** | 0.582 | 1.267 | −0.15 | 1.117 | 79 |
| Luxembourg | LUX | −0.003 | 0.004 | −0.004 | 0 | **160** | 0.001 | 0.066 | −0.037 | 0.029 | 127 |
| Macedonia, FYR | MKD | 0 | 0.051 | −0.027 | 0.024 | **146** | −0.145 | 0.552 | −0.509 | 0.043 | 150 |
| Madagascar | MDG | 1.057 | 0.756 | 0.557 | 1.313 | **23** | 6.589 | 7.034 | 2.105 | 9.138 | 33 |
| Malawi | MWI | 0.293 | 0.109 | 0.238 | 0.346 | **50** | 3.274 | 3.105 | 1.735 | 4.84 | 52 |
| Malaysia | MYS | 1.908 | 0.975 | 1.456 | 2.431 | **12** | 36.059 | 15.157 | 28.825 | 43.983 | 5 |
| Mali | MLI | 0.148 | 0.464 | −0.098 | 0.366 | **72** | 0.274 | 4.09 | −2.198 | 1.892 | 100 |
| Malta | MLT | −0.001 | 0.002 | −0.002 | 0 | **158** | −0.009 | 4.09 | −0.017 | 0 | 137 |
| Martinique | MTQ | 0.002 | 0.002 | 0.001 | 0.003 | **136** | −0.026 | 0.034 | −0.045 | −0.012 | 142 |
| Mauritania | MRT | −0.017 | 0.107 | −0.084 | 0.023 | **170** | 0.228 | 0.505 | −0.007 | 0.498 | 105 |
| Mexico | MEX | 1.366 | 1.904 | 0.467 | 2.37 | **19** | 21.758 | 44.215 | 2.513 | 46.727 | 13 |
| Micronesia, Fed. Sts. | FSM | −0.001 | 0.825 | −0.004 | 0 | **159** | −0.02 | 15.478 | −0.049 | 0 | 141 |
| Moldova | MDA | 0.043 | 0.051 | 0.016 | 0.067 | **99** | −0.38 | 0.867 | −0.791 | 0.076 | 156 |
| Mongolia | MNG | 0.273 | 0.335 | 0.132 | 0.466 | **52** | −2.415 | 4.261 | −4.838 | −0.577 | 171 |
| Montenegro | MNE | 0.002 | 0.007 | −0.002 | 0.006 | **138** | −0.275 | 0.353 | −0.462 | −0.109 | 155 |
| Morocco | MAR | 0.186 | 0.309 | 0.005 | 0.314 | **65** | 0.461 | 1.068 | −0.05 | 1.018 | 91 |
| Mozambique | MOZ | 0.719 | 0.326 | 0.601 | 0.927 | **30** | 10.785 | 6.795 | 7.422 | 14.217 | 24 |
| Myanmar | MMR | 1.822 | 0.486 | 1.576 | 2.062 | **15** | 33.651 | 7.463 | 29.817 | 37.28 | 6 |
| Namibia | NAM | 0.055 | 0.026 | 0.039 | 0.065 | **93** | 1.606 | 1.09 | 0.891 | 1.981 | 65 |
| Nepal | NPL | 0.268 | 0.15 | 0.218 | 0.368 | **53** | 1.304 | 4.198 | −0.606 | 3.592 | 68 |
| Netherlands | NLD | 0.037 | 0.026 | 0.028 | 0.054 | **104** | 1.218 | 0.487 | 0.989 | 1.476 | 71 |
| New Caledonia | NCL | 0.002 | 0.005 | 0 | 0.005 | **135** | −0.132 | 0.292 | −0.302 | −0.01 | 148 |
| New Zealand | NZL | 0.209 | 0.504 | −0.085 | 0.419 | **61** | −5.615 | 9.948 | −9.952 | −0.004 | 180 |
| Nicaragua | NIC | 0.595 | 0.625 | 0.256 | 0.881 | **37** | 5.482 | 7.275 | 1.926 | 9.202 | 36 |
| Niger | NER | 0.198 | 0.557 | −0.166 | 0.391 | **62** | 4.826 | 15.275 | −5.289 | 9.986 | 37 |





**Table A3.** Mean, range, minimum and maximum estimates of country-level annual mean (2011 – 2021) and cumulative (1950 – 2021) net $f_{\text{LULUCF}}$ as derived from three bookkeeping models, plus the respective country's rank. Countries are sorted in alphabetical order. Note, the rank of the cumulative $f_{\text{LULUCF}}$ (in bold) gives the order of countries as they occur in Figs. A9 – A18.

| Country | Code | Cumulative $f_{\text{LULUCF}}$ in 1950 – 2021 (GtC) | | | | | Annual mean $f_{\text{LULUCF}}$ in 2011 – 2021 (MtC yr$^{-1}$) | | | | |
| | | Mean | Range | Min | Max | **Rank** | Mean | Range | Min | Max | Rank |
|---|---|---|---|---|---|---|---|---|---|---|---|
| Nigeria | NGA | 2.235 | 1.716 | 1.452 | 3.168 | **8** | 6.813 | 9.342 | 1.421 | 10.763 | 32 |
| Niue | NIU | 0 | 0.625 | 0 | 0 | **151** | 0 | 7.275 | 0 | 0 | 130 |
| Norway | NOR | 0.045 | 0.096 | 0.012 | 0.109 | **97** | 0.533 | 1.736 | −0.265 | 1.472 | 83 |
| Oman | OMN | 0.001 | 0.002 | −0.001 | 0.002 | **143** | 0.043 | 0.085 | −0.011 | 0.074 | 113 |
| Pakistan | PAK | 0.303 | 0.478 | 0.119 | 0.597 | **49** | 3.539 | 3.92 | 1.953 | 5.873 | 47 |
| Panama | PAN | 0.214 | 0.187 | 0.11 | 0.298 | **59** | 1.891 | 0.265 | 1.758 | 2.023 | 61 |
| Papua New Guinea | PNG | 0.45 | 0.055 | 0.425 | 0.48 | **41** | 9.124 | 1.3 | 8.583 | 9.883 | 28 |
| Paraguay | PRY | 1.044 | 0.232 | 0.935 | 1.166 | **25** | 19.552 | 8.934 | 15.271 | 24.205 | 14 |
| Peru | PER | 1.046 | 0.586 | 0.762 | 1.348 | **24** | 12.15 | 15.182 | 4.111 | 19.293 | 21 |
| Philippines | PHL | 1.762 | 0.561 | 1.397 | 1.958 | **16** | 9.798 | 7.545 | 4.819 | 12.365 | 27 |
| Poland | POL | −0.703 | 1.034 | −1.162 | −0.128 | **184** | −11.805 | 15.326 | −17.92 | −2.594 | 185 |
| Portugal | PRT | −0.012 | 0.068 | −0.043 | 0.025 | **167** | −0.97 | 0.767 | −1.232 | −0.465 | 163 |
| Puerto Rico | PRI | −0.018 | 0.044 | −0.046 | −0.002 | **171** | −0.52 | 0.761 | −1.019 | −0.258 | 160 |
| Qatar | QAT | 0 | 0 | 0 | 0.001 | **148** | 0.003 | 8.934 | 0 | 0.005 | 124 |
| Romania | ROU | 0.037 | 0.364 | −0.141 | 0.223 | **105** | −2.985 | 3.122 | −4.21 | −1.088 | 175 |
| Russian Federation | RUS | 2.255 | 6.204 | −0.657 | 5.548 | **7** | 6.293 | 127.326 | −54.465 | 72.861 | 34 |
| Rwanda | RWA | 0.122 | 0.1 | 0.079 | 0.179 | **76** | 0.508 | 0.28 | 0.335 | 0.615 | 84 |
| Samoa | WSM | 0.003 | 0.004 | 0.001 | 0.005 | **131** | 0.018 | 0.136 | −0.062 | 0.075 | 117 |
| Saudi Arabia | SAU | 0.004 | 0.173 | −0.107 | 0.065 | **130** | 0.019 | 1.207 | −0.73 | 0.477 | 116 |
| Senegal | SEN | 0.08 | 0.048 | 0.056 | 0.104 | **85** | 0.305 | 2.816 | −1.17 | 1.646 | 98 |
| Serbia | SRB | −0.051 | 0.06 | −0.089 | −0.029 | **174** | −1.899 | 2.203 | −3.318 | −1.115 | 168 |
| Sierra Leone | SLE | 0.25 | 0.28 | 0.135 | 0.416 | **56** | 3.692 | 1.143 | 3.009 | 4.152 | 46 |
| Singapore | SGP | 0.003 | 0.006 | −0.001 | 0.005 | **132** | 0.076 | 0.085 | 0.028 | 0.113 | 110 |
| Slovak Republic | SVK | −0.027 | 0.171 | −0.125 | 0.045 | **172** | −0.582 | 1.706 | −1.525 | 0.182 | 161 |
| Slovenia | SVN | 0.015 | 0.027 | 0.001 | 0.029 | **122** | 0.503 | 0.81 | 0.088 | 0.898 | 85 |
| Solomon Islands | SLB | 0.026 | 0.036 | 0.009 | 0.045 | **111** | 0.894 | 1.226 | 0.328 | 1.555 | 72 |
| Somalia | SOM | 0.156 | 0.163 | 0.085 | 0.248 | **70** | 4.182 | 1.355 | 3.708 | 5.063 | 42 |
| South Africa | ZAF | 0.647 | 0.829 | 0.359 | 1.188 | **33** | 0.477 | 4.063 | −1.633 | 2.43 | 88 |
| South Sudan | SSD | 0.185 | 0.28 | 0.079 | 0.36 | **66** | 2.058 | 3.124 | 0.696 | 3.82 | 58 |
| Spain | ESP | −0.123 | 0.01 | −0.129 | −0.118 | **179** | −4.309 | 1.758 | −4.918 | −3.16 | 177 |
| Sri Lanka | LKA | 0.256 | 0.315 | 0.123 | 0.438 | **55** | 2.245 | 2.344 | 1.335 | 3.678 | 56 |
| St. Lucia | LCA | 0.001 | 0.002 | 0 | 0.002 | **142** | −0.015 | 0.015 | −0.025 | −0.01 | 139 |
| St. Vincent and the Grenadines | VCT | 0.001 | 0 | 0 | 0.001 | **144** | −0.002 | 0.01 | −0.008 | 0.002 | 132 |
| Sudan | SDN | 0.607 | 0.511 | 0.335 | 0.847 | **36** | 13.14 | 6.012 | 10.81 | 16.822 | 20 |
| Suriname | SUR | 0.037 | 0.028 | 0.024 | 0.052 | **106** | 0.857 | 0.201 | 0.768 | 0.969 | 73 |
| Swaziland | SWZ | 0.014 | 0.033 | 0.002 | 0.034 | **124** | 0.015 | 0.243 | −0.097 | 0.145 | 119 |
| Sweden | SWE | 0.04 | 0.244 | −0.117 | 0.127 | **101** | 3.306 | 5.622 | 1.004 | 6.625 | 50 |
| Switzerland | CHE | −0.015 | 0.06 | −0.055 | 0.005 | **169** | −0.389 | 0.879 | −0.921 | −0.042 | 157 |
| Syrian Arab Republic | SYR | 0.015 | 0.069 | −0.016 | 0.052 | **121** | −0.115 | 0.108 | −0.155 | −0.046 | 147 |
| Tajikistan | TJK | 0.041 | 0.081 | 0.013 | 0.095 | **100** | 0.304 | 1.105 | −0.09 | 1.015 | 99 |
| Tanzania | TZA | 1.984 | 1.241 | 1.35 | 2.591 | **11** | 42.549 | 37.438 | 30.02 | 67.458 | 4 |
| Thailand | THA | 2.203 | 2.188 | 1.421 | 3.609 | **10** | 16.437 | 18.813 | 9.827 | 28.64 | 16 |
| Timor-Leste | TLS | 0.05 | 0.059 | 0.02 | 0.079 | **94** | 0.736 | 1.008 | 0.123 | 1.131 | 77 |
| Togo | TGO | 0.069 | 0.099 | 0.019 | 0.119 | **90** | 0.683 | 1.297 | −0.067 | 1.23 | 78 |
| Tonga | TON | 0.001 | 0.001 | 0.001 | 0.002 | **139** | 0.011 | 0.008 | 0.007 | 0.015 | 121 |
| Trinidad and Tobago | TTO | −0.004 | 0.016 | −0.013 | 0.003 | **162** | −0.097 | 0.054 | −0.117 | −0.064 | 145 |
| Tunisia | TUN | 0.044 | 0.07 | 0.002 | 0.072 | **98** | 0.193 | 0.285 | 0.055 | 0.341 | 107 |
| Turkey | TUR | 0.406 | 0.671 | 0.165 | 0.836 | **44** | −4.481 | 9.471 | −8.859 | 0.612 | 179 |
| Turkmenistan | TKM | 0.183 | 0.473 | 0.022 | 0.496 | **67** | 0.744 | 2.457 | −0.142 | 2.315 | 76 |
| Uganda | UGA | 0.617 | 0.049 | 0.595 | 0.644 | **34** | 4.652 | 3.902 | 2.072 | 5.974 | 38 |
| Ukraine | UKR | 0.517 | 0.392 | 0.336 | 0.727 | **39** | 0.262 | 6.855 | −4.214 | 2.642 | 101 |
| United Arab Emirates | ARE | −0.006 | 0.016 | −0.015 | 0 | **164** | −0.138 | 0.318 | −0.345 | −0.027 | 149 |
| United Kingdom | GBR | −0.057 | 0.349 | −0.264 | 0.084 | **175** | 0.914 | 3.914 | −2.335 | 1.578 | 106 |
| United States | USA | −1.027 | 12.575 | −9.116 | 3.459 | **186** | −26.7 | 104.577 | −92.463 | 12.115 | 186 |
| Uruguay | URY | 0.022 | 0.13 | −0.049 | 0.081 | **112** | 1.83 | 4.515 | −1.041 | 3.475 | 62 |
| Uzbekistan | UZB | 0.145 | 0.315 | 0.038 | 0.353 | **73** | −0.213 | 2.985 | −1.474 | 1.512 | 152 |
| Vanuatu | VUT | 0.019 | 0.022 | 0.008 | 0.029 | **114** | −0.083 | 0.397 | −0.33 | 0.067 | 143 |
| Venezuela, RB | VEN | 0.974 | 0.606 | 0.694 | 1.3 | **27** | 10.628 | 11.413 | 3.756 | 15.169 | 25 |
| Vietnam | VNM | 1.302 | 1.102 | 0.697 | 1.799 | **21** | 29.403 | 36.544 | 9.856 | 46.4 | 8 |
| Western Sahara | ESH | −0.001 | 0.003 | −0.002 | 0 | **157** | 0.005 | 0.805 | 0 | 0.008 | 123 |
| Yemen, Rep. | YEM | 0.005 | 0.021 | −0.006 | 0.015 | **128** | −0.1 | 0.401 | −0.31 | 0.091 | 146 |
| Zambia | ZMB | 0.789 | 0.881 | 0.401 | 1.282 | **28** | 14.804 | 11.187 | 10.109 | 21.296 | 19 |
| European Union | EU | −3.009 | 7.570 | −7.053 | 0.517 | **–** | −50.232 | 114.972 | −108.633 | 2.268 | – |





*Author contributions.* JP and WAO designed the study. AB, CS, FNT, GC, GG, JP, RAH, SS, TG and WAO contributed to processing and evaluating the data. WAO led the analysis, and drafted the manuscript with contributions from all coauthors. Additionally, the help of Tobias Nützel is greatly acknowledged.

*Competing interests.* The authors declare that they have no conflict of interest, but point out that one author (FNT) is a thematic editor for the same journal.

*Disclaimer.* The views expressed in this manuscript are the authors' only and may not under any circumstances be regarded as stating an official position of the European Commission or the FAO.

*Acknowledgements.* We thank all people and institutions within the "Trends and drivers of the regional-scale sources and sinks of carbon dioxide" (TRENDY) modelling groups who contributed to the data used in this study. TG acknowledges support from the European Union's Horizon 2020 research and innovation programme under grant agreement no. 101003536 (ESM2025 project). We are grateful to national experts for their provision of official country data to FAO. FAOSTAT is produced and maintained by the FAO Statistics Division thanks to regular funding to FAO from member states.





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
