# Peer review of "Country-level estimates of gross and net carbon fluxes from land use, land-use change and forestry"

_Earth System Science Data, 2023_

## Author Response (AR1)

################################ **EC1** ################################

Reviewer 1 (RC1) has pointed out an important concern about the originality and quality of the dataset, which is planned to be released upon the paper's acceptance and publication. The dataset originates from model outputs, not from raw or pre-processed data, making it crucial to perform both qualitative and quantitative analyses to explain variations in the different estimates provided. In line with RC1's observations, focusing on Bookkeeping models would provide a more realistic framework for discussing the study's implications concerning Nationally Determined Contributions (NDCs) under the Paris Agreement, as well as the necessity for a more robust carbon monitoring system. I strongly recommend that the authors address these points, along with any other concerns raised by the reviewers, in a revised version of the manuscript. The amended paper should then be resubmitted for additional review.

Dear editor,

We are pleased to present a revised version of our manuscript "**Country-level estimates of gross and net carbon fluxes from land use, land-use change and forestry**". We have considered all the points raised by the reviewers and incorporated most of their and your suggestions. Note, in the verbatim quotations shown, those text passages that were newly added as part of the revision are underlined, and the line numbers given refer to the revised manuscript.

We have expanded (1) the qualitative as well as (2) the quantitative analyses on the causes for the variations in the different estimates at various places in the manuscript:
(1) To improve the qualitative assessment, we have added a brief description of the most recent updates of the HYDE land use forcing dataset and expanded the discussion of its influence on the various $f_{LULUCF}$ estimates. For details, please refer to our responses to the reviewers' comments (see response 2 to reviewer 1 and our response to reviewer 2). (2) To improve the quantitative analyses, we have added new information on the correlation statistics of the differences of the various modelling estimates, at various places where we present and discuss the respective results that are shown in Figures 2 and 3:

l. 288ff - "These consistent trends and peaks lead to comparably high correlation coefficients between the BLUE19 and present-day DGVM estimates, for example in Brazil, China and Nigeria. However, the correlation coefficients for the estimates from the different modelling approaches are generally rather low, which is due to the interannual variability, which is captured by the DGVMs but usually not by the BKs. In line with this, the particularly low correlation coefficients between DGVMs and BKs in the USA and Russia result from high interannual variability in combination with a low signal from land use changes." [here follows the example of Indonesia]

l. 307ff - "Large LUHv2 forcing differences, as indicated by particularly low correlation coefficients, are found in Brazil, DR Congo and Nigeria, with substantially larger land-use emission estimates using the more recent HYDE3.3 data."

l. 339ff - "What is striking here is the particularly low correlation coefficient in the USA. Similar to the difference between the modelling approaches, this results primarily from the

combination of a low land use signal with very high interannual variability in environmental conditions."

We agree that from today's perspective a focus on bookkeeping models seems to provide the most realistic framework for discussing NDCs and the need for a more robust carbon monitoring system. This is because inventory-based approaches are not capable of realistically deriving the necessary country contributions to stay within the remaining carbon budget for reaching the targets of the Paris agreement as inventory-based approaches (1) include fluxes from natural and indirect anthropogenic effects on all managed land and (2) rely on differing definitions and methods than bookkeeping models and Integrated Assessment Methods (compare e.g. Grassi et al., 2021 - doi:10.1038/s41558-021-01033-6). We have added a half sentence to highlight this mismatch in the manuscript (l. 183ff - "In addition, $f_{LULUCF}$ estimates from country reports rely on different definitions and assumptions than global models and therefore do not allow, for example, a realistic derivation of countries' climate change mitigation contributions that are consistent with the pathways to achieve the climate targets of the Paris Agreement as derived by Integrated Assessment Models."), which is followed by the description of the aforementioned reasons. In contrast, bookkeeping models can be used for quantifying the remaining carbon budget as they estimate $f_{LULUCF}$ without including fluxes due to natural and indirect anthropogenic effects on managed land. However, one important limitation is that bookkeeping models do not consider the environmental impact on $f_{LULUCF}$ (that is, for example, net biome productivity responses to elevated temperature, nitrogen and atmospheric $CO_2$), which Dynamic Global Vegetation Models (DGVMs) are capable of including. The large interannual variability in $f_{LULUCF}$ estimates from the DGVMs (compare Figs. 2 & 3) is a result of the inclusion of these natural and indirect anthropogenic effects. Noteworthy, this effect will become increasingly important in the future as various climate scenarios with different $CO_2$ concentrations and with different frequencies and intensities of climate extremes impact net biome productivity and thus $f_{LULUCF}$ estimates in different ways. As DGVMs are able to consider these environmental effects, their application will be key for assessing the impacts of different future emissions and climate scenarios, not least as DGVMs are integral parts of the widely used Earth System Models. Furthermore, the inclusion of DGVM output into our analysis provides the essential measures to improve comparison with, for example, numerous regional studies that use (in some cases exclusively) DGVM output for their analysis (e.g. Kondo et al., 2022; Rosan et al., 2021; Petrescu et al., 2022). We are therefore convinced that it is essential to use DGVMs for $f_{LULUCF}$ estimation and to compare the DGVM output with those from other assessment approaches. In addition, the usage of the DGVMs enables us to quantify the extent to which the environmental forcing and the land use forcing data influence $f_{LULUCF}$. To clarify these points, we have expanded the description of the importance of the environmental feedbacks, which was only possible by using the DGVM simulations from TRENDYv8 under different environmental forcings (compare also earlier comment and our response 3 to reviewer 1).

l. 353ff "It should be emphasised that the strong influence of environmental factors, which is reflected in the up- and downswings of the DGVM estimates, is likely to become more important under future climate conditions with more frequent and intense extreme environmental conditions, and potentially decreasing LULUCF activities as set out by the Glasgow Leaders' Declaration on Forests and Land Use."

Taken together, this study provides an unprecedented consolidated data set integrating $f_{LULUCF}$ estimates from DGVMs, bookkeeping models and inventory-based approaches enabling to analyze the uncertainties and differences of the various assessment approaches on the country-level.

Sincerely yours,

Wolfgang A. Obermeier (on behalf of all authors)

############################# **RC1** #############################

We thank the reviewer for the time and effort spent for this thorough revision. We are pleased to present a revised and improved version of our manuscript, in which we have taken into account all of the reviewer's comments and incorporated most of the suggestions. Note, in the verbatim quotations shown, those text passages that were newly added as part of the revision are underlined, and the line numbers given refer to the revised manuscript.

[Comment 1] This manuscript compared the country-level gross and net carbon fluxes from land use, land-use change and forestry globally, and the estimates come from Global Carbon Budget project, FAO and National Greenhouse Gas Inventories (NGHGIs). Basically, this topic is important to understand the global carbon budget. However, this study did not produce any new datasets, and all datasets used in this study come from the existing datasets. The dataset only available in this manuscript is gross fluxes of land cover and change from three Book-keeping models. Although in the global carbon budget (GCB), the gross fluxes should be submitted but they are not available in the website of GCB. However, these gross fluxes from three BK models are not enough to make this manuscript be published as the journal 'encourage submissions on original data or data collections which are of sufficient quality and have potential to contribute to these aims'.

[Response 1] Indeed, as stated in the manuscript, country-level estimates from bookkeeping models, UNFCCC, and FAOSTAT have already been published. In contrast, country-level net $f_{LULUCF}$ estimates from DGVMs, namely from TRENDYv8 simulations under present-day and transient environmental forcings as well as transient TRENDYv11 simulations have not been published elsewhere. To the best of our knowledge, country-level data from DGVMs are so far only available for forest fluxes on managed land and on unmanaged land that are used by Grassi et al. (2023) to adjust the NGHGI data to the definition used in bookkeeping models (as also done in our study). In addition to the country-level net flux DGVM data, we publish country-level gross flux data from bookkeeping models, which have not yet been publicly available. These data are becoming increasingly important, not least to independently assess the two levers for combating climate change (emission reductions and increasing removal). Taken together, this study provides the first country-level estimates of net $f_{LULUCF}$ from DGVMs as well as gross $f_{LULUCF}$ from bookkeeping models, and provides the first consolidated compilation of the different datasets, allowing for a comprehensive comparison. We are therefore convinced that the data provided in this study adds substantially to the aims of *Earth System Science Data*.

[Comment 2] In addition, the manuscript compared the differences of gross and net carbon fluxes from land cover and change at country-level. However, most of these results from BK and DGVMs are well-known and reported by the previous papers, especially in the GCB papers during recent several years. I would like to suggest the authors analyzing the cause for differences of various estimates. The land cover dataset is one of the most important sources, especially for GCB, FAO and NGHGIs. The differences between BK models and DGVMs majorly originate from model structure. The latter is quite complicate topic, which may be not good to discuss in this manuscript, but I would like to see the former issue discussed at least.

[Response 2] We thank the reviewer for this comment and agree that the land use forcing data is of substantial relevance for estimating fluxes from LULUCF. In line with a comment from reviewer 2, this comment led us to expand the description of the $f_{LULUCF}$ differences resulting from differing land use forcing data as well as those resulting from the modelling approach. In addition, we have improved the description of the differences in the definitions from the various assessment methods, as can be seen in the response to the next comment. For the improved description of the impact of the land use forcing difference have therefore added a brief description of the updates in HYDE3.3 compared to HYDE3.2 and expanded the discussion of the impacts of the different land use forcing data on the individual countries at various places in the text.

I. 165ff - "The main innovations in HYDE3.3 were the provision of yearly output from 1950 onwards, the update of the onset of agriculture based on new radiocarbon data and archaeological expertise indicating more spatial heterogeneity, the use of the latest satellite data with increased spatial resolution on an annual basis from 1992 - 2018 from the European Space Agency (ESA) and MapBiomas data for Brazil for the period 1985 - 2020, as well as the inclusion of more sub-national cropland and pasture data."

I. 307ff - "Large LUHv2 forcing differences, as indicated by particularly low correlation coefficients, are found in Brazil, DR Congo and Nigeria, with substantially larger estimates using the more recent HYDE3.3 data."

I. 309ff - "The particularly big differences for Brazil mainly result from an improved representation of deforestation patterns through the inclusion of MapBiomas land cover data in the newer HYDE version, and for DR Congo, from the inclusion of revised data from the FAO (Friedlingstein et al., 2022). In China, for example, the H&N22 land use forcing data assumes a steady increase in forest areas from 1950, while the LUH2 data shows decreasing forest areas until 1990 and relatively stable forest areas thereafter (Yu et al., 2022)."

Moreover, we have expanded the discussion on the impacts of the environmental forcing and of the modelling approach (compare also our response to the editor).

I. 288ff - "These consistent trends and peaks lead to comparably high correlation coefficients between the BLUE19 and present-day DGVM estimates, for example in Brazil, China and Nigeria. However, the correlation coefficients for the estimates from the different modelling approaches are generally rather low, which is due to the interannual variability, which is captured by the DGVMs but usually not by the BKs. In line with this, the particularly low correlation coefficients between DGVMs and BKs in the USA and Russia result from high interannual variability in combination with a low signal from land use changes." [here follows the example of Indonesia]

I. 339ff - "What is striking here is the particularly low correlation coefficient in the USA. Similar to the difference between the modelling approaches, this results primarily from the combination of a low land use signal with very high interannual variability in environmental conditions."

[Comment 3] The third, several figures in this manuscript can be excluded because they did not express effective information to readers. Such as fig. A9-18 and Table A1-A3. The

readers can easily draw these figures based on the datasets provided by the manuscript. It will be better to represent more analyses about the cause of estimate differences. And I also would like to know the reason for mismatch between simulated fluxes and NGHGIs as the fig. 4 represented.

[Response 3] The reviewer is correct that these figures can be drawn based on the datasets provided by the manuscript. However, in reality, few readers will go through this effort. We see it as a service to the community to provide these figures, in a common and easily accessible format. We further believe that showing country-level data so comprehensively will stir constructive criticism at high detail and thus provides the basis for progress on more robust data.

We are grateful for the comment regarding the mismatch between simulated fluxes and NGHGIs (Fig. 4). The main reasons for the differences between the reported fluxes to the UNFCCC (NGHGIs) and the simulated fluxes are (1) the inclusion of indirect effects on managed land (e.g. due to increased $CO_2$ concentrations) in the reported fluxes, and (2) the greater area of land assumed to be managed in the country reports. To increase comparability of the different assessment approaches, we additionally plotted the "adjusted NGHGI DB" flux, which we have derived according to the approach from Grassi et al. (2023 - doi:10.5194/essd-15-1093-2023). Considering your comment and to highlight these factors more prominently, we have clarified the description of the approach for the adjusted NGHGI DB fluxes and extended the paragraph where the underlying reasons for the differences are discussed in the results section.

l. 130ff - "To improve comparability of the NGHGI DB data with modelled estimates, we adjust the NGHGI DB data to better match the processes and definitions of the modelled estimates (the basis in this study) by correcting for this so-called managed land issue (hereafter, adjusted NGHGI DB). Following the approach described in in Grassi et al. (2023), we therefore subtract from the NGHGI DB estimates those fluxes resulting from natural and indirect effects (i.e., human-induced environmental change) from managed land."

l. 386ff - "Much of this discrepancy (globally adding up to ~1.6 GtC yr$^{-1}$ for the period 2001 - 2020) can be explained by different definitions, particularly regarding two points (compare appendix and Grassi et al., 2023; Schwingshackl et al., 2022): (1) The inclusion of natural and indirect human-induced fluxes, such as those resulting from increased forest regrowth due to higher atmospheric $CO_2$ concentration and N deposition, in many country reports. (2) The larger area assumed to be managed land in the country reports compared to the BKs, which results, in combination with the inclusion of natural and indirect human-induced fluxes, in lower emissions reported by countries."

In addition, we have added a reference to the section where the adjustment approach of the NGHGI DB is described to the section describing the differences across the approaches.

l. 194 - "In order to make the NGHGI and BK land use flux estimates comparable we here translate NGHGI estimates by removing the fluxes that models attribute to the natural land sink (compare the adjusted NGHGI DB data described in Sect. 2.3)"

[Comment 4] The authors showed the comparison between BK models from GCB-2022 (v11) and DGVMs of Trendy-v8 mostly, and provided the Trendy-v11 results in the data file.

GCB used LUH2 dataset to indicate the global land cover and change, and LUH2 dataset was updated annually. There are large differences especially at tropical regions between GCB-LUH2 V8 and V11. Therefore, the estimates of BK-v8 and DGVMs-v11 should not be compared because different LUH2 version. It should be interesting to compare the land cover differences derived from different LUH2 versions.

[Response 4] We thank the reviewer also for this point. Indeed, the land-use forcing data used for the BK simulations for GCB2022 and used by the DGVM simulations from TRENDY-v8 differ, as you pointed out. This makes a direct comparison challenging. However, there exists no alternative to estimate the impact of the environmental forcing compared to the different land use forcing data for the latest GCB estimates, as simulations with present-day environmental forcing were last carried out under the TRENDY-v8 protocol (i.e. for GCB2019). We have also compared the estimates of the BK model BLUE from GCB2019 (BLUE19, using TRENDY-v8 forcing) with the GCB2022 estimates of BLUE (BLUE22, using TRENDY-v11 forcing). As the model code was not changed between these versions, this allowed us to isolate the impact of the different LUH2 versions (which we name "LUHv2 forcing difference"), i.e., excluding differences resulting from differing environmental forcings or model changes. In line with our response to your second comment and a comment from reviewer 2, we have expanded the description of the different land use data (namely, HYDE3.3 and HYDE3.2) and extended the discussion of their influences on modelled $f_{\text{LULUCF}}$ estimates (see our earlier responses).

This is a very comprehensive overview of CO$_2$ fluxes from LULUCF based on several databases. Comparing fluxes between so many different types of models could lead to a very complicated paper, but the authors have been very clear throughout the paper. The result is a very readable paper that highlights major differences, areas of agreement, and potential sources of uncertainty. I think the paper should be accepted with very little revision, I have some suggestions to improve the clarity in some places.

We thank the reviewer for the effort and time spent for the review as well as for the positive evaluation of our manuscript. In particular, we are pleased that the reviewer finds our compilation and analyses of datasets of CO$_2$ fluxes from LULUCF very clear. We have addressed all of the reviewer's comments and incorporated the suggested changes, which has improved the revised version of the manuscript, as described below. Note, in the verbatim quotations shown, those text passages that were newly added as part of the revision are underlined, and the line numbers given refer to the revised manuscript.

Page 3, Line 77: It would be helpful to include the values for the net f$_{LULUCF}$ here, too.

Thank you, we have added the net flux values. The text now reads as follows (l. 75ff): "[…] with the most recent GCB2022 estimating global anthropogenic gross emissions at 3.8 ± 0.7 GtC yr$^{-1}$ and gross removals at 2.6 ± 0.4 GtC yr$^{-1}$ for 2012 - 2021, being thus 2 - 4 times larger than the global net $f_{LULUCF}$ of 1.2 ± 0.7 GtC yr$^{-1}$ in this period (Friedlingstein et al., 2022b)."

Page 6, Line 163: The differences between the HYDE3.2 and HYDE3.3 datasets come up in Section 4.2. I suggest adding a short summary of the key differences between the datasets here to give some context to those later results.

Thank you for this point, which was also raised by reviewer 1. We have added a short description of the updates in the newer dataset HYDE3.3 based on a very recent data description of HYDE3.3 (https://doi.org/10.24416/UU01-67UHB4) and a presentation by Kees Klein Goldewijk within a recent workshop of the Global Carbon Project. Moreover, we have expanded our discussion on the impacts of the updated land use forcing data (compare response 2 to reviewer 1).

l. 165ff - "The main innovations in HYDE3.3 were the provision of yearly output from 1950 onwards, the update of the onset of agriculture based on new radiocarbon data and archaeological expertise indicating more spatial heterogeneity, the use of the latest satellite data with increased spatial resolution on an annual basis from 1992 - 2018 from the European Space Agency (ESA) and MapBiomas data for Brazil for the period 1985 - 2020, as well as the inclusion of more sub-national cropland and pasture data."

Page 8, Lines 225-233: The first and last sentence of this paragraph are essentially the same.

Thank you for spotting this mistake. We have deleted the second occurrence of the sentence an slightly reworded the first occurrence, which now reads as follows: "In the main

manuscript we exemplarily focus on the top eight countries with highest cumulative net LULUCF emissions in the period 1950 - 2021 based on BKs, as well as the USA, which have the highest cumulative removals in this period."

Table 1: Can you clarify how the standard deviation is taken. Is this the deviation across model estimates?

Yes, this is the deviation across the multiple model estimates. We have added this fact to the table's caption (Table 1): "The table indicates the mean $f_{LULUCF}$ estimates, their standard deviation across the different model estimates (SD), and their relative uncertainty (SD divided by the absolute mean value)."

Page 12, Lines 284-285: Why don't the DGVMs include the emissions from organic soils during the 1997 El Nino year?

In line with the GCB assessments, we have used the direct DGVM output without adding emissions from organic soils. In the GCB, this is a pragmatic approach that has little impact on the net LULUCF flux statistics, as the DGVMs are only used to assess uncertainty. Adding essentially the same peat emission estimates from the same external dataset as done for BKs does hardly affect the statistics shown in Figure 2.

Page 12, Line 288: Do you mean BLUE19 here: "To investigate this further, we compare the BLUE22 estimates with the 2022 estimates of all BKs …"

Thank you for this point. No, the sentence indeed refers to BLUE22 output, which is compared to the output from the other BKs in 2022 to show that, in most cases, BLUE estimates higher carbon fluxes upon land-use change due to higher differences in carbon densities between land-use types as compared to some other models. However, we understand that in connection with the previous sentence, this could be interpreted erroneously and therefore, we slightly reworded the sentence, which now reads as follows:

l. 298ff - "To investigate this further, we compare the estimates of all three BKs from 2022 (BKs 2022; note that BLUE model code was not changed between 2019 and 2022 versions)."

In addition, we have slightly altered the wording where we describe the often higher $f_{LULUCF}$ estimates from the BLUE model as compared to those from the other bookkeeping models:

l. 302ff - "This can mainly be explained by rather high differences in the carbon densities between natural and managed areas assumed in the BLUE model in conjunction with the fact the BLUE captures the full extent of (LUH2-based) gross transitions (compare Sect. 4.3 and description of the BKs in Sect. A1, and Bastos et al., 2021)."

l. 375ff - "This is mainly due to different assumed carbon densities, which, depending on the ecosystem, are rather high in the BLUE model and low in the H&N model (compare appendix and Bastos et al., 2021a, b)."

This manuscript describes estimates of CO2 fluxes from land-use, land-use change, and forestry over the period 1950-2021 at country-level spatial scales. These estimates are computed by book-keeping models (BKs), Dynamic Global Vegetation Models (DGVMs), and official country reports and inventories. The analysis includes comparisons between the various approaches and the resulting uncertainties in land-use carbon fluxes. Detailed results are given for 9 countries (the 8 countries with the largest cumulative emissions since 1950, as well as the USA). The authors explore the underlying reasons for differences in land-use emissions from each model and methodology.

This is a really useful and interesting investigation because land-use emissions have been very large historically and are likely to continue to be important in the future when the land is used for climate mitigation efforts. Land-use emissions also have a large amount of uncertainty associated with them and quantifying and reducing this uncertainty is an important activity for the carbon cycle community. This manuscript does a great job of documenting the various sources of uncertainty and the various approaches to estimating land-use-based carbon fluxes and providing these details all in one place. This will be a really useful resource for researchers who want to make steps towards reducing these uncertainties and providing the best estimates of recent carbon fluxes.

We thank the reviewer for this positive evaluation and for the efforts and time spent for the review of our manuscript. In particular, we are pleased about the finding that this manuscript does a great job of documenting the uncertainties in the various approaches to estimating LULUCF fluxes and provides steps to reduce these uncertainties. We considered all of the reviewer's comments and adopted the suggested changes, improving the revised version of the manuscript as described below. Note, in the verbatim quotations shown, those text passages that were newly added as part of the revision are underlined, and the line numbers given refer to the revised manuscript.

I think this manuscript is in good shape and ready for publication. I only have a couple of small suggestions:

1. It would be good to mention somewhere that the large land-use emissions estimated for some countries such as Brazil and Indonesia could possibly be driven by international trade demands from other countries. I realize that disentangling the drivers of land-use change is beyond the scope of this study, but I think it would be useful to add this as a caveat somewhere.

We thank the reviewer for this comment. This is a very important point, especially when it comes to equitable burden sharing and the development of assessment expectations regarding countries' contributions to climate change mitigation. So far, we had mentioned this fact in a short statement discussing the fact that the ten largest $f_{LULUCF}$ emitters, if calculated per area or per capita, are all located in the tropics (l. 252f - "However, most of these emissions are embodied in trade and are caused by consumption in industrialized regions such as Europe, the United States, and China (Hong et al., 2022)."). To make this point even more prominent, we have added a statement, in which we introduce the top eight emitters and highlight their potential for climate change mitigation:

l. 253ff - "However, it should be noted that the resulting call for action is not limited to these top emitters, as large parts of national LULUCF emissions, particularly in the tropics, are caused by consumption elsewhere (Hong et al., 2022)."

2. The authors mention (in the Conclusions) that the use of Earth observation data could provide substantial improvements in the estimation of land-use carbon fluxes and it would be great to get some more detail about this. For example, are there specific land-use component fluxes that will be most improved, or specific regions/countries? Are there specific datasets or modeling approaches that are most promising for this? Do they have any suggestions for how to incorporate this data into the suite of existing approaches (i.e. BKs, DGVMs, and country-reports)?

We also thank the reviewer for these important questions. From our perspective the most important land use components that could be improved within the modelling realm by the use of Earth observation data are the forest sink (particularly, by improving regrowth fluxes via incorporation of forest age classes from remote sensing and improved process representation of forest management), deforestation and forest degradation fluxes (by better spatially explicit data from Earth observations and processes related to degradation). The specific regions/countries where modelling approaches need to be improved by Earth observations should encompass, in particular, those areas located in carbon-rich biomes where a lot of LUC happen. Such efforts have already been successfully implemented for Brazil (starting with the GCB2021) and for Indonesia (starting with the GCB2023) and should be extended, for example for the DR Congo.

For the country-reports, we believe that particular improvement is needed in spatially explicit reporting of $f_{LULUCF}$ which could be enabled by incorporating remote sensing data on land use activities, and would provide the means to better compare the country reports with spatially explicit modelling estimates, even beyond the country-level. Additionally, Earth observations could strongly improve national GHG reporting in Non-Annex I countries, where data sources are scarce.

We have highlighted the potential for improvements via Earth observation data in the final sentences in Sect. 5 as follows:

l. 552ff - "In addition, all models (BKs and DGVMs) should use more spatially explicit forcing datasets. In particular, Earth observation data may provide improved spatially explicit data (e.g., of land degradation and restoration efforts), on carbon densities in vegetation and soil, forest regrowth by incorporation of forest age classes and forest management from optical satellite measurements, as well as data on carbon fluxes from atmospheric inversions. Furthermore, spatially explicit data on land use activities from Earth observations could improve the quality of report-based estimates where data sources are scarce, and improve comparability with estimates from modelling approaches. Such an improved incorporation of Earth observation data into modeling and country report-based approaches may provide substantial advancements in the assessment and understanding of $CO_2$ fluxes from LULUCF."

3. It would be great if the authors could provide a short list of what they consider to be the most important next steps for reducing the uncertainty in country-level land-use carbon

fluxes. In other words, which are the largest "uncertainty hotspots" that the land-use carbon flux community should attempt to address first?

We thank the reviewer for this important point, which has prompted us to expand the description and discussion of the main sources of uncertainty and ways to reduce these uncertainties (see also response to the last comment). Despite the respective extensions in the text, we would like to emphasize that the topic of uncertainties and future improvements of $f_{LULUCF}$ estimates is, in our opinion, so comprehensive that it constitutes a topic for a separate paper.

l. 538ff - "For example, the uncertain emission estimates from BKs in India, from the 1980s onward in China, and for the most recent decades in Brazil and DR Congo could be improved by better land use forcing data. Likewise, the uncertain removal estimates in Russia and India could be improved by incorporating better land use forcing data and improved process representation in models, such as for fluxes related to land abandonment.

l. 544ff - "For example, in Canada and Nigeria, increasing differences between modelled estimates and those from the NGHGI DB, even after adjustment of the latter, call for in-depth analysis of the underlying drivers."

l. 548ff - "In addition, the still existing definition and framework issues implicitly underlying all datasets, e.g. definition/inclusion of LASC, transient C densities or not, Biome and PFT definitions, managed land proxy, etc., should be addressed. To further increase the confidence in $f_{LULUCF}$ estimates, more approaches similar to bookkeeping models, for example by DGVM simulations under a BK-like protocol, could be used."